# Using light and X-ray scattering to untangle complex neuronal orientations and validate diffusion MRI

**Miriam Menzel[1,2]\*‡, David Gräßel[2], Ivan Rajkovic[3], Michael M Zeineh[4]†, Marios Georgiadis[4]\*†**

[1]Department of Imaging Physics, Faculty of Applied Sciences, Delft University of Technology, Delft, Netherlands; [2]Institute of Neuroscience and Medicine (INM-1), Forschungszentrum Jülich GmbH, Jülich, Germany; [3]Stanford Synchrotron Radiation Lightsource, SLAC National Accelerator Laboratory, Standford, United States; [4]Department of Radiology, Stanford School of Medicine, Stanford, United States

**Abstract** Disentangling human brain connectivity requires an accurate description of nerve fiber trajectories, unveiled via detailed mapping of axonal orientations. However, this is challenging because axons can cross one another on a micrometer scale. Diffusion magnetic resonance imaging (dMRI) can be used to infer axonal connectivity because it is sensitive to axonal alignment, but it has limited spatial resolution and specificity. Scattered light imaging (SLI) and small-angle X-ray scattering (SAXS) reveal axonal orientations with microscopic resolution and high specificity, respectively. Here, we apply both scattering techniques on the same samples and cross-validate them, laying the groundwork for ground-truth axonal orientation imaging and validating dMRI. We evaluate brain regions that include unidirectional and crossing fibers in human and vervet monkey brain sections. SLI and SAXS quantitatively agree regarding in-plane fiber orientations including crossings, while dMRI agrees in the majority of voxels with small discrepancies. We further use SAXS and dMRI to confirm theoretical predictions regarding SLI determination of through-plane fiber orientations. Scattered light and X-ray imaging can provide quantitative micrometer 3D fiber orientations with high resolution and specificity, facilitating detailed investigations of complex fiber architecture in the animal and human brain.

**\*For correspondence:**
m.menzel@tudelft.nl (MM);
mariosg@stanford.edu (MG)

†These authors contributed equally to this work

**Present address:** ‡Department of Imaging Physics, Faculty of Applied Sciences, Delft University of Technology, Delft, Netherlands

**Competing interest:** The authors declare that no competing interests exist.

## Editor's evaluation

This paper presents a valuable cross-validation study of mesoscopic measurements of axonal orientations from three different modalities – small-angle X-ray scattering, scattered light imaging, and diffusion MRI – and is of interest to researchers who want to apply these methods to white matter imaging. The authors show convincing similarities and differences in fiber orientations from all three methods over partial ex vivo brain samples, though the validation of diffusion MRI could be strengthened by considering other fiber reconstruction methods, as only a single diffusion method is investigated.

## Introduction

Unraveling the complex nerve fiber network in the brain is key to understanding its function and alterations in neurological diseases. The detailed reconstruction of multiple crossing, long-range nerve fiber pathways in densely-packed white matter regions poses a particular challenge. *Diffusion magnetic resonance imaging* (dMRI) is currently used to derive axonal orientations in vivo. However, with voxel

**Figure 1.** Comparison of X-ray scattering (top) and light scattering (bottom) for analyzing nerve fiber structures. (**A**) Principle of X-ray scattering on a nerve fiber bundle, resulting in a ring at a radial position corresponding to myelin's periodicity, strongest perpendicular to the in-plane fiber orientation. (**B**) Principle of light scattering on a nerve fiber bundle, which similarly yields scattered photons perpendicular to the in-plane fiber orientation. (**C**) Schematic drawing of a 3D-scanning small-angle X-ray scattering (3D-sSAXS) measurement of a brain section, in which raster scanning at multiple angles allows reconstructing 3D fiber orientation distributions for each point of illumination. (**D**) Schematic drawing of a scattered light imaging (SLI) scatterometry measurement of a whole brain section (left) and reconstruction of a scattering pattern for one selected image pixel (right), which can be performed over the entire image simultaneously.

sizes typically down to a few hundred micrometers in post-mortem human brains (*Calabrese et al., 2018*; *Roebroeck et al., 2019*), the resolution is insufficient to resolve individual nerve fibers, and isolating the anisotropic signal coming from myelinated axons alone is challenging. Moreover, up to hundreds of fibers within a voxel might have complicated geometries, e.g., crossing or kissing fibers, which poses a further challenge.

*Small-angle X-ray scattering* (SAXS) provides myelinated nerve fiber orientations by studying the anisotropy of myelin diffraction (Bragg) peaks in X-ray scattering patterns (*Figure 1A and C – Georgiadis et al., 2020*; *Georgiadis et al., 2021*). These are generated by the interaction of the incoming X-ray photons with the layered structure of the myelin sheath, which surrounds nerve fibers in the white matter. The method can be tomographic (SAXS tensor tomography – *Liebi et al., 2015*; *Schilling et al., 2019a*; *Georgiadis et al., 2021*), and 3D-scanning SAXS (3D-sSAXS) can provide 3D distributions of axon orientations in tissue sections (*Georgiadis et al., 2020*). Recent studies further revealed that SAXS can exploit the modulations in the azimuthal position of the myelin-specific Bragg peaks to resolve crossing nerve fiber populations across species with resolutions of tens of micrometers (*Georgiadis et al., 2023*).

The scattering of visible light can also be used to reveal crossing nerve fiber orientations (*Figure 1B – Menzel et al., 2020a*; *Menzel and Pereira, 2020b*) as it is sensitive to directional arrangements of myelinated axons (~µm diameter). In *scattered light imaging* (SLI) (*Menzel et al., 2020a*; *Menzel et al., 2021b*; *Reuter and Menzel, 2020*) the sample (brain section) is illuminated from many different angles and a camera captures an image of the brain section (*Figure 1D*, left), in which the intensities of each image pixel vary with the angle of illumination. In this way, a scattering pattern is generated

for each micron-sized image pixel (*Figure 1D*, right). SLI has been shown to reliably reconstruct up to three in-plane fiber orientations for each image pixel (with an accuracy of +/−2.4°; *Menzel et al., 2021a*) and an in-plane resolution at the micrometer scale (*Menzel et al., 2021b*).

Hence, a combined measurement of 3D-sSAXS and SLI, with the high specificity to myelinated fibers of the former and the high-resolution capabilities of the latter, can potentially provide a gold standard for imaging complex nerve fiber orientations in thin brain sections with micrometer resolution.

Here, we present 3D-sSAXS and SLI measurements on the same tissue samples (coronal sections from vervet monkey and human brains) and compare them to dMRI outcomes. To capture multiple possible fiber scenarios, we examine brain regions with both unidirectional and crossing fibers – the corpus callosum and corona radiata, respectively. Evaluation of 3D-sSAXS and SLI in a vervet brain section provides a unique cross-validation, but also a very detailed mapping of the single and crossing fiber orientations. Comparison of the results on the human brain sample enables validation of dMRI-derived orientations. Furthermore, we enhance the interpretation of out-of-plane fibers in SLI, using the 3D-fiber orientations from SAXS and dMRI as references. The presented SAXS and SLI results on the same specimens and their cross-validation provide the groundwork for an eventual combined analysis of the two techniques, exploiting the specificity of the former and the resolution of the latter. This combined information could be used to provide reliable nerve fiber orientations and validate dMRI results, towards more accurate brain connectivity maps of the animal and human brain.

## Results
### Light and X-ray scattering patterns reflect various nerve fiber configurations

To better understand how light and X-ray scattering patterns correspond to each other for different nerve fiber configurations, we analyzed the scattering patterns from SLI and SAXS measurements in a vervet monkey brain section. *Figure 2* shows the resulting scattering patterns for four representative points (marked with asterisks in B): (i) a unidirectional in-plane fiber bundle in the corpus callosum, (ii) two crossing fiber bundles in the corona radiata, (iii) a slightly through-plane inclined fiber bundle in the fornix, and (iv) a steep out-of-plane fiber bundle in the cingulum. The orientation information is encoded in the variation of the signal intensity as a function of the azimuthal angle $\varphi$ (going in a circle around the pattern, *Figure 2C* (i)), plotted as *azimuthal profile* under each scattering pattern in *Figure 2C*. *Figure 2—figure supplement 1* shows the average, maximum, minimum, mean peak prominence, and mean peak width of the azimuthal profiles for each pixel measured with SAXS and SLI.

While the SLI scattering patterns show contiguous signal intensity (from center out), the strongest SAXS signal (Bragg peaks) appears along the Debye-Scherrer ring (arrows in *Figure 2C*), at a specific distance (q-value) from the center of the pattern that corresponds to the myelin layer periodicity (here 17.5 nm) (*Georgiadis et al., 2021*).

For in-plane nerve fibers, i.e., nerve fibers that mostly lie within the section plane, the strongest signal in both SLI and SAXS is perpendicular to the fiber orientation (red dashed lines in *Figure 2B and C* (i)), shown in the azimuthal profile as peaks lying 180° apart. For the two in-plane crossing fiber bundles in the corona radiata (ii), the peaks in the SLI and SAXS azimuthal profiles similarly indicate the fiber orientations, with each bundle producing two peaks lying 180° apart (white/yellow arrows). In the following, the term 'peak' will be used to refer to peaks in azimuthal profiles, so that a pair of azimuthal 'peaks' always corresponds to a single fiber orientation.

For partly out-of-plane fibers, i.e., fibers that have a certain angle with respect to the section plane, such as those in the fornix, the peaks in the SAXS azimuthal profiles are still 180° apart - owing to the center-symmetry of the pattern -, but become less pronounced with increasing out-of-plane fiber angle (compare peak height of SAXS, *Figure 2C* (i) vs. (iii)). In contrast, the between-peak distance in the SLI azimuthal profiles decreases with increasing fiber inclination (SLI, *Figure 2C* (iii)), as also predicted by simulation studies (*Menzel et al., 2020a*). For out-of-plane fibers that run almost perpendicular to the section plane (*Figure 2* point (iv), cingulum), both the SLI scattering pattern and the SAXS myelin ring become almost radially symmetric without significant peaks in the azimuthal profile. In such cases, the information about the in-plane fiber orientations is limited, whereas the out-of-plane angle can

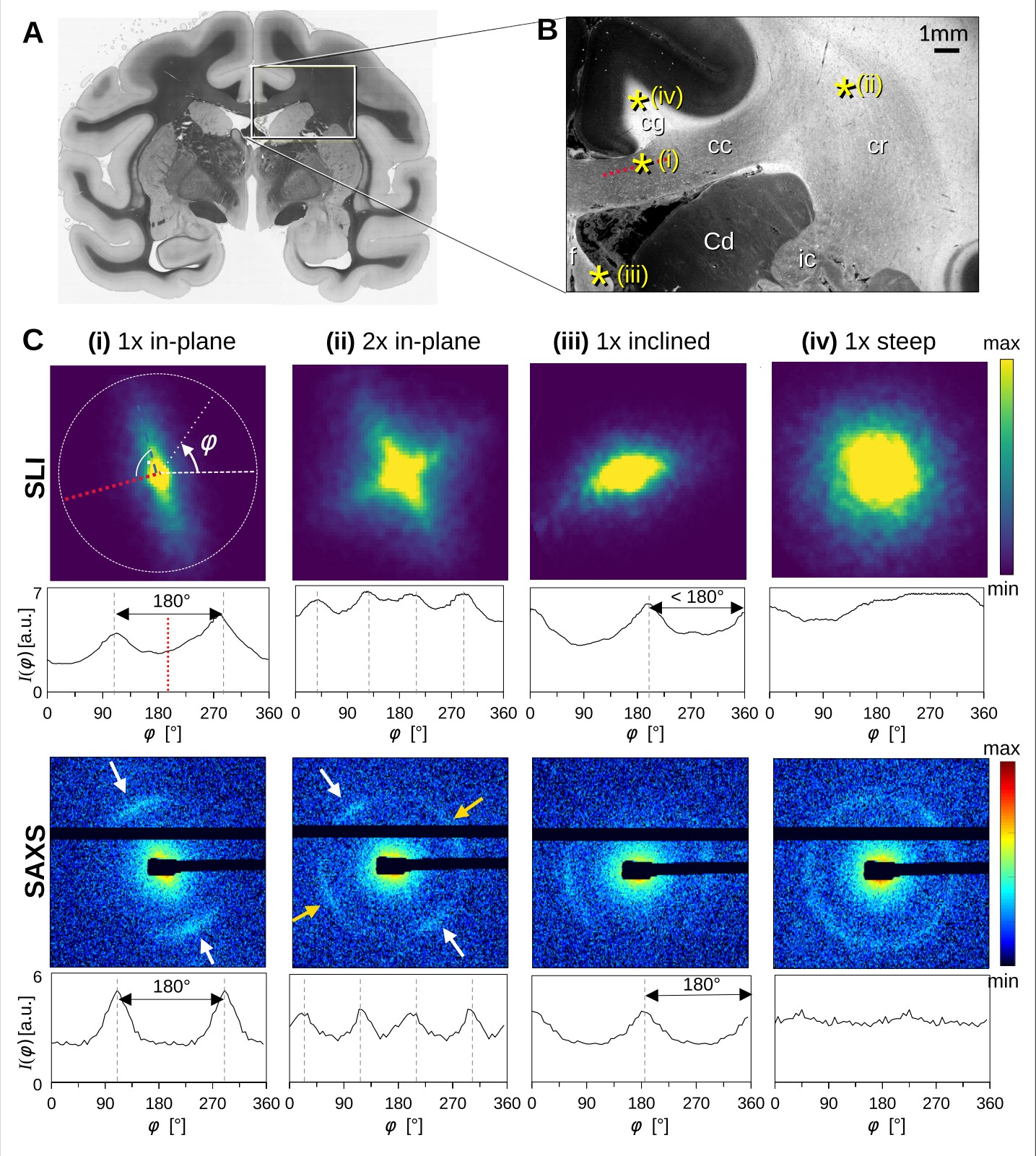

**Figure 2.** Scattering patterns obtained from scattered light imaging (SLI) scatterometry (px = 3 μm) and small-angle X-ray scattering (SAXS) (px = 100 μm) on a 60 μm-thick vervet monkey brain section at a coronal plane between the amygdala and hippocampus (section no. 511). (**A**) Transmittance image of the whole brain section. (**B**) Average scattered light intensity of the investigated region (cc: corpus callosum, cr: corona radiata, cg: cingulum, Cd: caudate nucleus, f: fornix, ic: internal capsule). Yellow asterisks indicate the points corresponding to the scattering patterns in **C**.

*Figure 2 continued on next page*

*Figure 2 continued*

(**C**) Scattering patterns from SLI (top) and SAXS (bottom) with azimuthal profiles plotted beneath each pattern, obtained from the pixels indicated in **B**. (i) unidirectional in-plane fiber bundle in the corpus callosum, with peaks perpendicular to the fiber orientation (red dotted line), lying 180° apart, (ii) two in-plane crossing fiber bundles in the corona radiata, (iii) slightly inclined fiber bundle in the fornix, with SLI peaks <180° apart, and SAXS peaks 180° apart but with lower Bragg peak intensity, (iv) highly inclined fiber bundle in the cingulum, with no SLI azimuthal peak pair and almost isotropic SAXS myelin ring.

The online version of this article includes the following figure supplement(s) for figure 2:

**Figure supplement 1.** Parameter maps obtained from small-angle X-ray scattering (SAXS) and scattered light imaging (SLI) azimuthal profiles for vervet brain section no. 511.

be retrieved with scans at different rotation angles using 3D-sSAXS (*Georgiadis et al., 2020*), and approximated in SLI (*Menzel et al., 2021b*).

## SAXS and SLI resolve crossing fibers and show high inter-method reproducibility

We then sought to more precisely compare the in-plane nerve fiber orientations derived from the peak positions in the SAXS and SLI azimuthal profiles, examining the same ~1 × 2 cm² region of the vervet brain (*Figure 3* and *Figure 3—figure supplement 1*). Given the ~33 x higher resolution of SLI versus SAXS in the presented measurements (3 µm vs. 100 µm pixels), smaller nerve fiber bundles e.g., in the head of the caudate nucleus (yellow arrow) can be traced in detail by SLI. Conversely, out-of-plane nerve fibers in the cingulum (cg), are more sensitively depicted by SAXS.

The in-plane nerve fiber orientations are highly coincident, not only for unidirectional fibers, but also for fiber crossings, depicted as colored lines in *Figure 3B*, *Figure 3—figure supplement 1D*, where each vector glyph covers orientations from a grid of 165 × 165 measured pixels that are visually overlaid in SLI, vs. a 5 × 5 pixel grid in SAXS. Further zooming in shows a concordant fiber course in the highly complex corona radiata architecture (*Figure 3C*, *Figure 3—figure supplement 1E*): the fibers of the corpus callosum fan out (blue/magenta) while crossing the ascending internal/external capsule (green).

To quantitatively compare the in-plane fiber orientations, SAXS images were linearly registered onto SLI images (scaled, rotated, and translated), and pixels in which both techniques yield the same number of fiber orientations (one or two) were compared to each other: For each image pixel, the corresponding fiber orientations were subtracted (SLI – SAXS), taking the minimum of the two possible pairings in regions with crossing fibers (*Figure 4*). *Figure 4C* shows the image pixels for which both techniques yield a single fiber orientation (magenta) or two fiber orientations (green). The quality of registration is shown in *Figure 4—figure supplement 1A*: The boundaries of white and gray matter are overall well aligned, only the fornix is slightly shifted (see arrows) and was, therefore, evaluated separately.

*Figure 4A* shows that the small positive and negative angular differences (displayed in shades of red and blue, respectively) appear to be randomly distributed across the tissue. The absolute angular differences are displayed in *Figure 4B*. While in-plane and slightly inclined fibers (corpus callosum and fornix) as well as major parts of crossing fibers in the corona radiata show mostly differences less than 10°, highly inclined fibers in the cingulum and the corona radiata show absolute differences of 20° and more (white arrows). The distribution of angular differences in white matter pixels with one and two fiber orientations is shown in *Figure 4D* (histograms in magenta and green, respectively). The two histograms show a distribution around zero degrees (one fiber orientation: mean ~0.017°, median absolute ~4.1°; two fiber orientations: mean ~0.316°, median absolute ~5.6°). While regions with one fiber orientation yield differences between +/−30° maximum, regions with two fiber orientations show multiple outliers with differences of +/−45° and more. As 33 × 33 SLI pixels with different fiber orientations correspond to one SAXS pixel with a single fiber orientation, larger differences between in-plane fiber orientations are expected, especially in regions with highly varying fiber orientations.

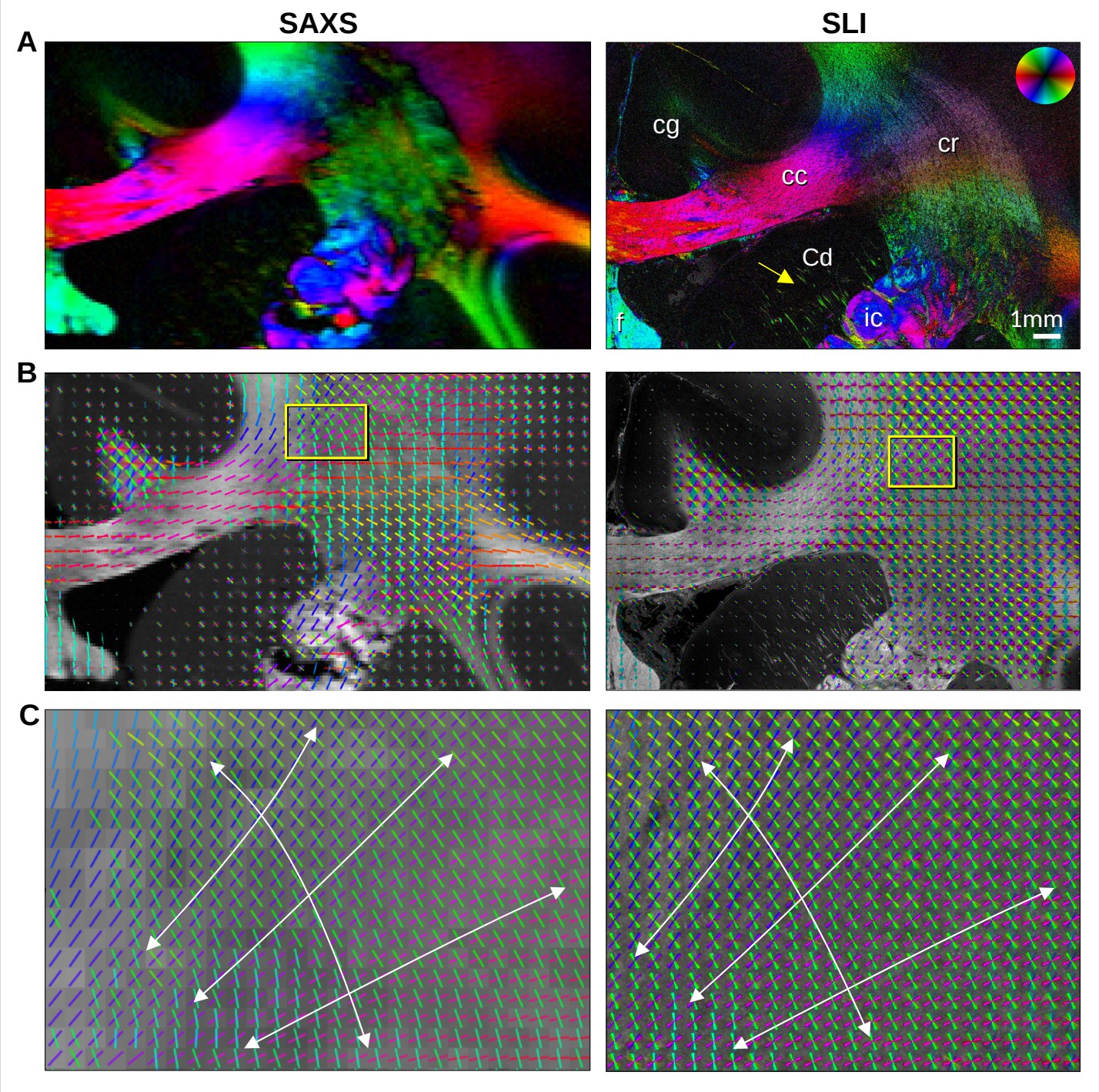

**Figure 3.** In-plane nerve fiber orientations from small-angle X-ray scattering (SAXS) and scattered light imaging (SLI) measurements of vervet monkey brain section no. 511. (**A**) Fiber orientation maps showing the predominant fiber orientation for each image pixel in different colors (see the color wheel in the upper right corner): px = 100 µm (SAXS), px = 3 µm (SLI). (cc: corpus callosum, cr: corona radiata, cg: cingulum, Cd: caudate nucleus, f: fornix, ic: internal capsule). (**B**) Fiber orientations are displayed as colored lines for 5 × 5 px (SAXS) and 165 × 165 px (SLI) superimposed. The length of the lines is weighted by the averaged scattered light intensity in SAXS and SLI, respectively. (**C**) Enlarged region of the corona radiata, showing fiber orientations as colored lines for 1 × 1 px (SAXS) and 33 × 33 px (SLI) superimposed. The white arrows indicate the main stream of the computed fiber orientations.

The online version of this article includes the following figure supplement(s) for figure 3:

**Figure supplement 1.** In-plane fiber orientations from small-angle X-ray scattering (SAXS) and scattered light imaging (SLI) measurements of vervet brain section no. 501.

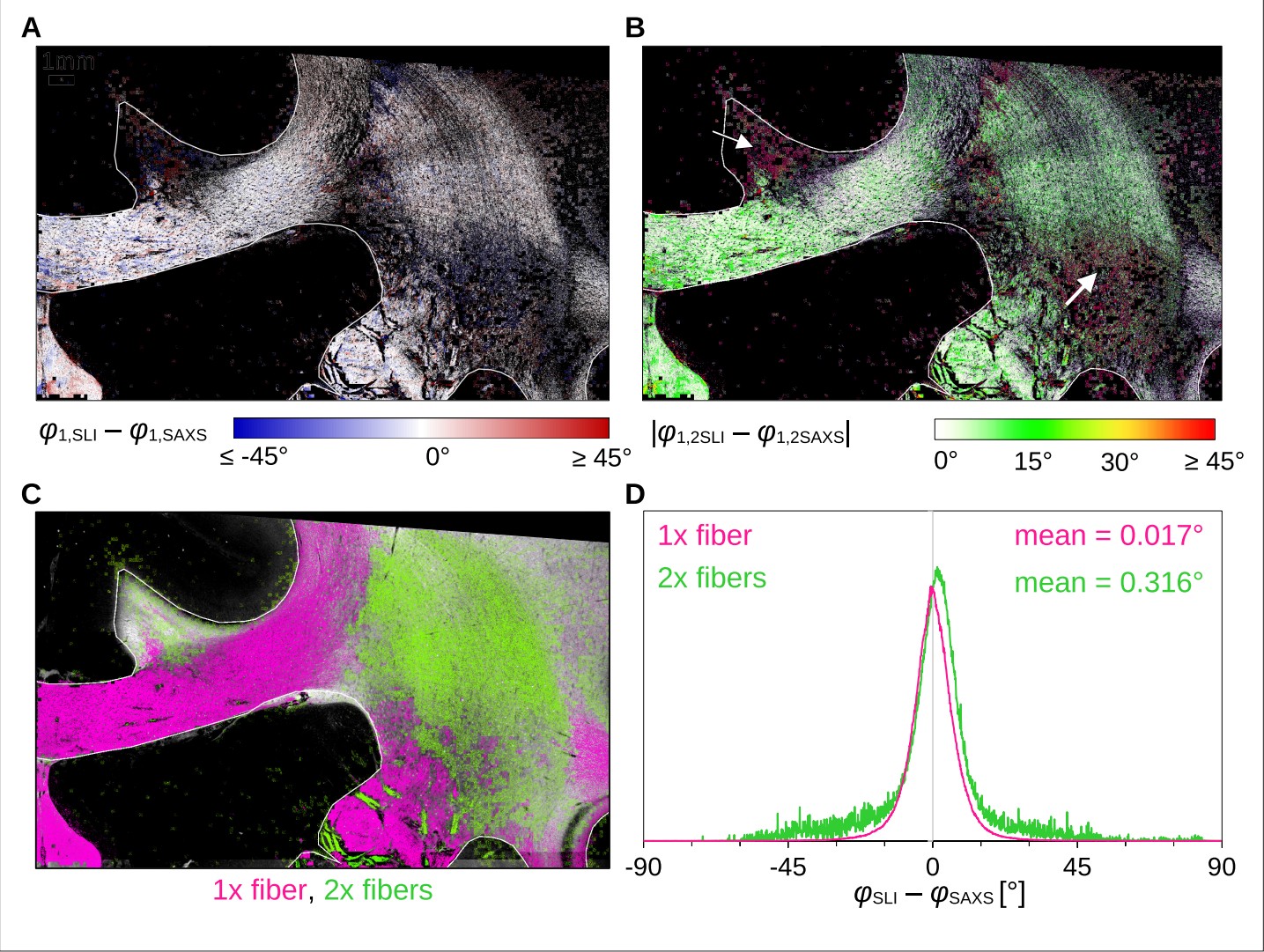

**Figure 4.** Angular difference between scattered light imaging (SLI) and small-angle X-ray scattering (SAXS) nerve fiber orientations (SLI – SAXS) for vervet monkey brain section no. 511. For evaluation, the SAXS image was registered onto the SLI image (with 3 μm pixel size) and only regions where both techniques yield the same number of fiber orientations were considered. (**A**) Angular difference displayed for one of the maximum two predominating fiber orientations in each pixel. (**B**) Absolute angular difference displayed for each image pixel. (**C**) Regions with one or two fiber orientations for both methods (magenta = 1, green = 2 orientations). (**D**) Histograms showing the angular difference for pixels with one and two fiber orientations, evaluated in white matter regions excluding the fornix (see regions delineated by white lines in **A-C**).

The online version of this article includes the following figure supplement(s) for figure 4:

**Figure supplement 1.** Quality of registration for vervet and human brain samples.

## Comparison to diffusion MRI shows high agreement and small discrepancies

Next, we aimed to extend our findings to the human brain and compare our results to diffusion MRI (dMRI) fiber orientations. To enable the analysis of regions with both unidirectional and crossing fibers, we selected a ca. 1 cm thick human brain sample that contains parts of the corpus callosum (cc), the cingulum (cg), the internal capsule (ic), and the corona radiata (cr) (*Figure 5A*). After high-resolution multi-shell dMRI scanning, we computed traditional diffusion tensors to yield the main fiber orientations (*Figure 5C*, left), and used multi-shell multi-tissue constrained spherical deconvolution (*Jeurissen et al., 2014*) to map fiber orientation distribution (*Figure 5D*, *Figure 5—figure supplement 1*), including regions with highly aligned fibers as well as distinct fiber crossings.

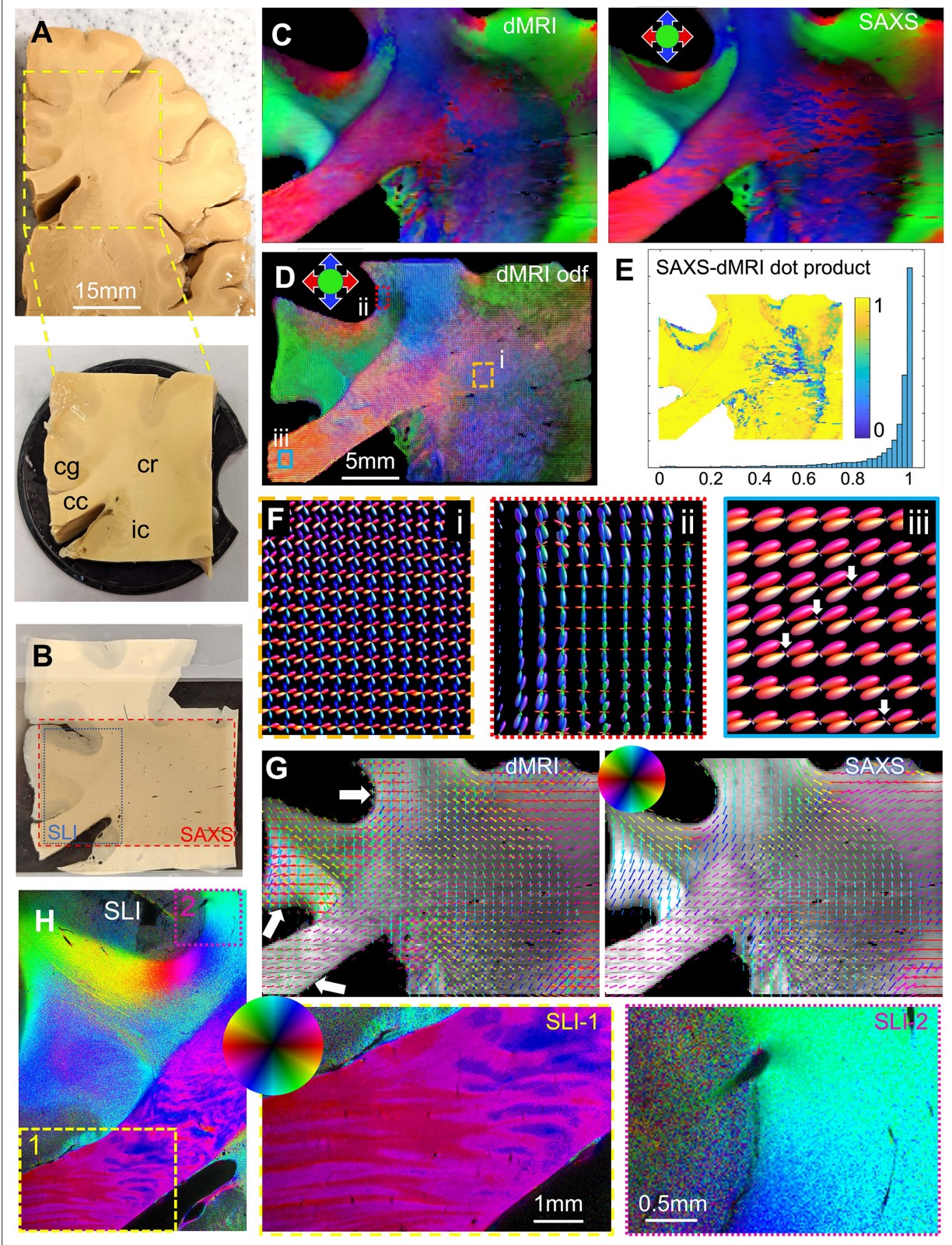

**Figure 5.** Diffusion magnetic resonance imaging (dMRI) measurement of a 3.5 × 3.5 × 1 cm³ human brain specimen (200 µm voxel size) in comparison to measurements with 3D-scanning SAXS (3D-sSAXS) (150 µm pixel size) and scattered light imaging (SLI) (3 µm pixel size) of a 80 µm-thick brain section. (**A**) Human brain specimen; the bottom image shows the sample measured with dMRI (cc: corpus callosum, cg: cingulum, cr: corona radiata, ic: internal capsule). (**B**) Posterior brain section with regions measured by 3D-sSAXS (red rectangle) and SLI (blue rectangle). (**C**) Main 3D fiber orientations from

*Figure 5 continued on next page*

*Figure 5 continued*

dMRI (left) and 3D-sSAXS (right) for the brain section; dMRI was registered onto 3D-sSAXS (with 150 µm pixel size), see *Figure 4—figure supplement 1B*. (**D**) Orientation distribution functions from dMRI, with zoomed-in regions surrounded by rectangles shown in subfigure **F**. (**E**) Vector dot product of the dMRI and 3D-sSAXS main fiber orientations, as histogram and map of the studied area. (**F**) The enlarged regions from subfigure **D** show the fiber orientation distributions in the corona radiata (rectangle i, orange), a subcortical fiber bundle (rectangle ii, red), and the corpus callosum (rectangle iii, cyan). The fiber orientation distributions are almost identical to the ones shown here also when only high b-values are taken into account, *Figure 5—figure supplement 4*. (**G**) In-plane fiber orientation vectors for dMRI (left) and SAXS (right) superimposed on mean SAXS intensity. Vectors of 5 × 5 pixels are overlaid to enhance visibility. Zoomed-in images of the corona radiata region from both methods are shown in *Figure 5—figure supplement 3C*. (**H**) In-plane fiber orientations from SLI (multiple fiber orientations are displayed as multi-colored pixels), with zoomed-in areas in boxes (1) and (2). For better readability, fiber orientations in the gray matter are not shown in subfigures **C-G**.

The online version of this article includes the following figure supplement(s) for figure 5:

**Figure supplement 1.** Anatomic (b0 – T2w) and diffusion MRI-based metrics for the human brain sample.

**Figure supplement 2.** Small-angle X-ray scattering (SAXS)-diffusion magnetic resonance imaging (dMRI) comparison for the anterior human brain section no. 20.

**Figure supplement 3.** Quantifying in-plane angular differences between small-angle X-ray scattering (SAXS) and diffusion magnetic resonance imaging (dMRI) for the corpus callosum (cc) and corona radiata (cr) areas of the posterior human brain section no. 18.

**Figure supplement 4.** MRI fiber orientations of the posterior human brain section no. 18, calculated from all b-values and high b-values only (high b-values: 5 and 10 ms/µm$^2$), with enlarged boxes i, ii, iii at the bottom.

**Figure supplement 5.** Single and crossing small-angle X-ray scattering (SAXS)- and diffusion magnetic resonance imaging (dMRI)-derived fiber orientations per voxel for the posterior human brain section no. 18.

To compare dMRI- with SAXS-derived fiber orientations, we measured two 80 µm-thick vibratome sections (one from the posterior side, *Figure 5*, and one from the anterior side, *Figure 5—figure supplement 2*) with 3D-scanning SAXS and computed fiber orientation distributions (*Georgiadis et al., 2015*, *Georgiadis et al., 2020*). To enable a quantitative comparison of the 3D fiber orientations obtained from dMRI and 3D-sSAXS, the dMRI sections corresponding to the physical SAXS-scanned sections (*Figure 5B*, red rectangle) were identified, and linearly registered to the SAXS data sets. *Figure 4—figure supplement 1B* shows the quality of registration: The main features, including the tissue and white/gray-matter boundaries, appear to be overall well aligned. The main fiber orientations per pixel for dMRI and 3D-sSAXS (*Figure 5C*) show a high correlation, similar to what has been shown in *Georgiadis et al., 2020*, with a dot product approximating unity (*Figure 5E*), and a median angular difference of 14.4° over all voxels (9.1° over voxels with fractional anisotropy (FA) >0.2, computed from the eigenvalues of the rank-2 tensor for both methods).

We then performed a more detailed analysis including crossing fibers. First, in the challenging region of the corona radiata, where multiple fiber crossings occur, the dMRI fiber orientations seem to be in high agreement with the direct structural X-ray scattering (*Figure 5G* and *Figure 5—figure supplement 3B* left): the two methods have a median angular difference of 5.6° in the first orientation, and 6.0° in the second orientation (overall median angular difference 5.8°). This shows that diffusion MRI has the sensitivity to accurately resolve multiple fiber orientations per voxel in the white matter, see also *Figure 5D and F* (rectangle i, orange in the corona radiata).

Next, we turned our focus to areas that appear to have relatively homogeneous fiber populations in SAXS, such as the corpus callosum. The main fiber orientations in these regions were again in high agreement between the two methods, with a median angular difference of 5.7° (*Figure 5—figure supplement 3B*, right). There was an observed difference in detecting secondary orientations in areas such as the corpus callosum. Unlike X-ray scattering, which shows homogeneous fiber orientations, dMRI seems to show multiple fiber orientations per voxel, with a secondary fiber population perpendicular to the main one (albeit with a much smaller magnitude). This is exemplified in the corpus callosum and in the subcortical white matter nearby the cingulate and the callosal sulci (white arrows in *Figure 5G and F*). The effect is also visible when calculating fiber orientations using the high b-values only (see *Figure 5—figure supplement 4*). Referencing these regions in the micrometer-resolution SLI (px = 3 µm, *Figure 5H*), we confirm the X-ray scattering results and do not observe a second fiber population perpendicular to the main one.

We then proceeded to quantify this effect over the entire white matter of the posterior brain section. Comparing the SAXS and dMRI secondary orientations, we observed a 104% (more than double) increase in the voxels with multiple orientations in dMRI. More specifically, secondary fiber

orientations within a single voxel were detected in 31% of the total number of voxels by SAXS vs. 64% of the total number of voxels by dMRI (also see *Figure 5—figure supplement 5* for a map of the number of orientations per voxel). The anterior brain section similarly showed a 40% increase, while primary orientations were in high agreement as well (*Figure 5—figure supplement 2*).

### Experimental validation of out-of-plane fiber orientations in SLI

While SLI determines the in-plane fiber orientation with high precision, out-of-plane fiber orientation (inclination) is challenging. Theory suggests that the fiber inclination is directly related to the distance between the two peaks in the SLI azimuthal profile (upper *Figure 2C*). The peak distance should decrease with increasing inclination, as indicated by the black dashed curves in *Figure 6G*, which were computed from simulated SLI azimuthal profiles for fiber bundles with different inclinations (*Menzel et al., 2021a*, Figure 7d). Measurements of SLI and 3D-sSAXS on the same tissue sample enable testing of this prediction, given the very high agreement of 3D-sSAXS and dMRI in the human brain sample in regions of out-of-plane fibers (*Figure 5C–E*).

We performed a pixel-wise comparison of the out-of-plane fiber orientation angles $\alpha$ from 3D-sSAXS (*Figure 6A and B*) and the peak distances $\Delta$ from SLI (*Figure 6D and E*), for one vervet brain section (A, D) and one human brain section (B, E). The quality of co-registration is shown in *Figure 4—figure supplement 1*. The 3D-arrows in *Figure 6A* indicate the orientation in which the nerve fibers point out of the section plane, computed by 3D-sSAXS for four selected regions. The images in *Figure 6C and F* show the corresponding 3D fiber orientations from the dMRI measurement of the human brain sample for reference.

The out-of-plane inclination angles from dMRI (*Figure 6C*) highly agree with those obtained from 3D-sSAXS (*Figure 6B*). In both coronal brain sections (vervet and human), the fibers in the corpus callosum (cc) are mostly oriented in-plane (dark blue: $\alpha<20°$), while fibers in the cingulum (cg) are mostly oriented out-of-plane (light green/yellow: $\alpha>40°$). Fibers in the vervet fornix (*Figure 6A*) show mostly intermediate inclination angles (light blue: $20°<\alpha<40°$).

When comparing the inclination angles to the corresponding SLI peak distances in *Figure 6D–E* (evaluated for regions with a single detected fiber orientation), it becomes apparent that regions with in-plane fibers (cc) contain many image pixels with large peak distances (blue: $\Delta>170°$), whereas regions with out-of-plane fibers (cg) contain many image pixels with notably smaller peak distances (green/yellow: $\Delta<140°$) – especially in the human cingulum. To quantify this effect, we plotted the SLI peak distances against the corresponding 3D-sSAXS inclinations for all evaluated image pixels (see scatter plots in *Figure 6G*; data points are shown in similar colors as the corresponding outlines in *Figure 6D-E*; the insets show the representative SLI azimuthal profiles and corresponding peak distances alongside the black dashed-line theoretical prediction).

The scatter plots confirm a decreasing peak distance with increasing fiber inclination for most regions, matching the prediction by simulations. The broadly distributed points from the cingulum might be due to the fact that the peak distance in regions with highly inclined fibers is harder to determine due to less pronounced peaks (*Figure 2C* (iv)). The data points in the white matter of the human cingulate gyrus (CiG) differ the most from the theoretically predicted curve (brown data points in *Figure 6G*) implying a stronger heterogeneity of the fiber architecture: While SAXS yields similarly high fiber inclinations as in the cingulum (magenta data points), the SLI peak distances are much larger (mostly between 160–180°). The large number of gray pixels (surrounded by a brown outline in *Figure 6E*) indicates the existence of crossing fibers. The dMRI orientation distribution functions (*Figure 6F*) reveal indeed that – in addition to the cingulum bundle with highly inclined fibers (in green) – the cingulate gyrus is interspersed with a transverse rather in-plane fiber bundle (in red), which explains the large SLI peak distances in some regions of the white matter cingulate gyrus.

### Discussion

We performed SLI and SAXS measurements on the same vervet monkey and human brain sections and compared our human section results to high-resolution ex vivo dMRI measurements of the same sample. This allowed us to cross-validate the techniques and to identify possible limitations. SAXS and SLI highly agree in areas of both single and crossing fiber orientations. Taking the main out-of-plane fiber orientations from dMRI and SAXS into account, we could show that SLI provides information

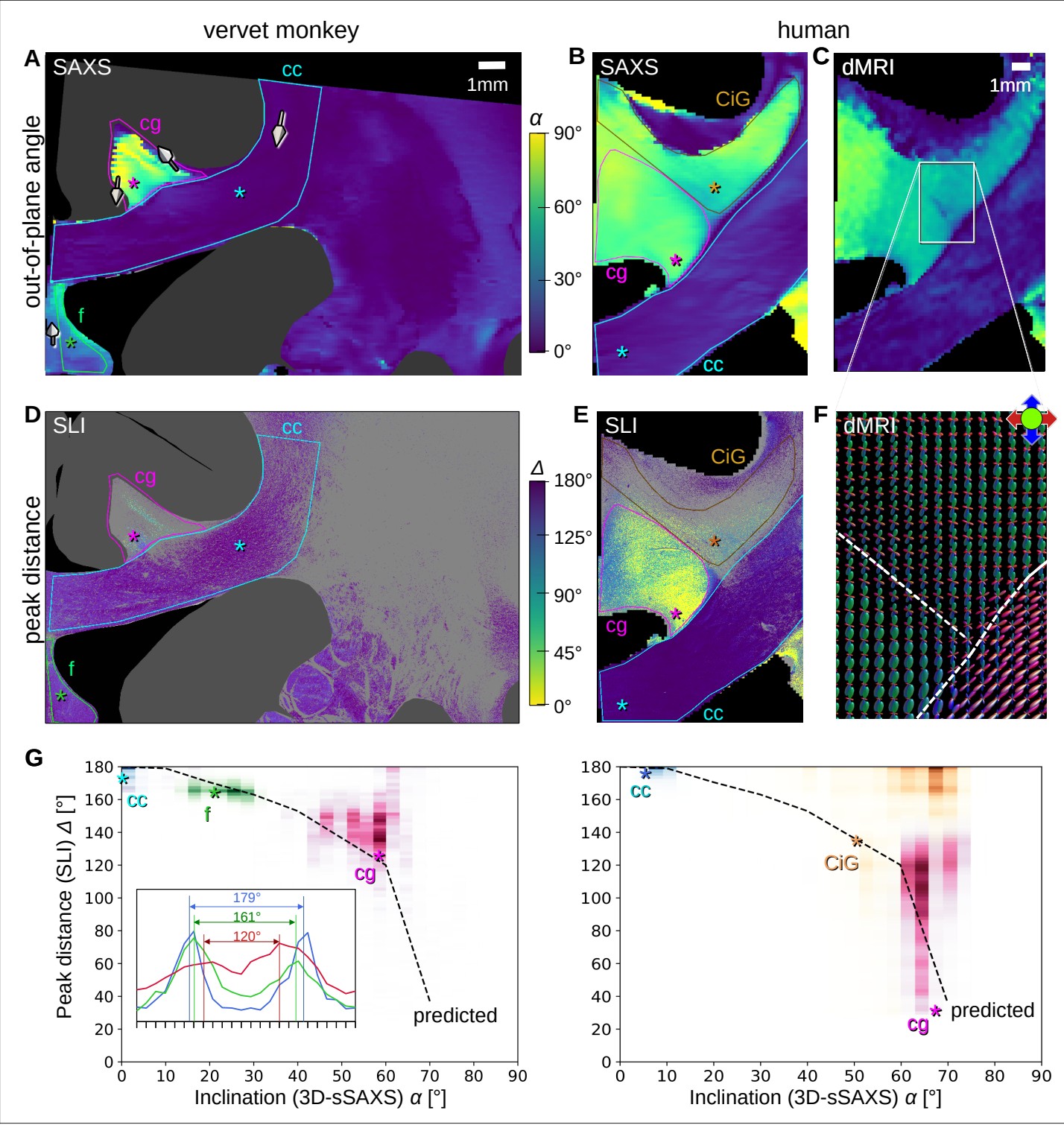

**Figure 6.** Pixel-wise comparison of 3D-scanning SAXS (3D-sSAXS)/diffusion magnetic resonance imaging (dMRI) fiber inclinations and scattered light imaging (SLI) peak distances. The images on the left show the analysis of one vervet brain section (no. 511, *Figure 2B*); the images on the right show the analysis for one human brain section (posterior section, cf. blue rectangle in *Figure 5B*). 3D-sSAXS and dMRI images were registered onto the corresponding SLI images with 3 μm resolution (*Figure 4—figure supplement 1*); the fornix in the vervet brain section was additionally shifted between the SLI and 3D-sSAXS images to account for the slight misalignment between the registered images in this region; only regions with unidirectional fibers were evaluated. (**A,B**) 3D-sSAXS fiber inclination angles of the vervet and human brain section are shown in different colors for the white matter (blue: in-plane, yellow: out-of-plane, dark gray: gray matter). The 3D-arrows indicate in four selected regions of the vervet brain section the orientation in which

*Figure 6 continued on next page*

*Figure 6 continued*

the fibers of these regions point out of the section plane. (**C**) Corresponding dMRI fiber inclination angles of the human brain sample. (**D,E**) Distance between two peaks in the corresponding SLI azimuthal profiles (cf. inset in **G**). Only profiles with one or two azimuthal peaks were evaluated (other pixels are shown in light gray). Regions used for the pixel-wise comparison with 3D-sSAXS are surrounded by colored outlines; asterisks mark three representative pixels. (**F**) dMRI orientation distribution functions of the region marked in **C**. The dashed lines indicate separation into the three regions in **E** (cg – cingulum, CiG – cingulate gyrus, cc - callosum). (**G**) SLI peak distance plotted against the 3D-sSAXS inclination for the corresponding regions marked in **D** and **E** for vervet and human brain samples, respectively (data points are displayed in similar colors as the corresponding outlines in **D** and **E**). The inset in the left panel shows the SLI azimuthal profiles for the three representative pixels in the vervet brain section (marked by similarly colored asterisks in **D** and **G**). The SLI profiles were centered for better comparison; the ticks on the inset x-axis denote azimuth steps of 15°. The dashed curves indicate the predicted SLI peak distance obtained from simulated scattering patterns of fiber bundles with different inclinations (*Menzel et al., 2021a*, Figure 7d).

about 3D-fiber orientations, but is still limited in the quantification of the out-of-plane angles, especially in regions with crossing fibers. The combined SAXS and SLI measurements, with the unbiased resolving power of SAXS and the high-resolution power of SLI show that both methods can provide a reliable reference for axonal orientation maps, towards a gold standard to which other techniques such as dMRI can be compared.

## Existing methods to identify fiber orientations and tracts

A large variety of neuroimaging techniques exists to study nerve fiber architectures in the postmortem brain. Some techniques (just as SLI and 2D-SAXS) analyze thin tissue sections to assess brain tissue structures. Histological staining allows to study nerve fiber organizations with fine structural detail (*Amunts et al., 2013*; *Carriel et al., 2017*), but has limitations in white matter regions with densely packed nerve fibers. Structure-tensor analysis of Nissl-stained histology slides can reveal glial cell orientation along axons (*Schurr and Mezer, 2021*), but the resulting fiber orientations are in 2D; also, dehydration during staining can lead to tissue deformation. Serial electron microscopy (*Eberle and Zeidler, 2018*; *Salo et al., 2021*) enables the analysis of brain tissue structures at the highest detail, but is only feasible for very small sample sizes and also requires a complex and specific sample preparation, preventing the reuse of samples for other purposes.

To assess microscopic fiber structures in 3D volumes (without sectioning), tissue clearing followed by labeling and fluorescence microscopy imaging is commonly used. In recent years, it has served as validation for dMRI data (*Morawski et al., 2018*; *Stolp et al., 2018*; *Goubran et al., 2019*; *Leuze et al., 2021*). However, the clearing process causes tissue deformation (*Leuze et al., 2017*). Other methods to study nerve fiber structures in 3D and microscopic detail (without clearing) are two-photon fluorescence microscopy (*Zong et al., 2017*; *Costantini et al., 2021*), or optical coherence tomography (*Magnain et al., 2015*; *Men et al., 2016*) which relies on the back-scattering of light from a tissue block and images the superficial tissue layer before sectioning.

All previously mentioned methods require a directional analysis of the microscopic image data to extract orientation information (structure tensor analysis – *Khan et al., 2015*; *Wang et al., 2015*). For this purpose, a kernel including several neighboring image pixels is used, which limits the resolution; also, in regions with densely packed nerve fibers, intensity gradients are low, limiting analysis.

To directly obtain the axonal orientations, optical coherence tomography can be combined with polarized light (PS-OCT), which exploits the optical anisotropy (birefringence) of myelinated axons to determine their orientations (*Wang et al., 2018*; *Jones et al., 2020*; *Jones et al., 2021*). A similar principle is used in polarization microscopy where polarized light is passed through thin brain sections and alterations in the polarization state are measured – a technique known for more than a century (*Brodmann, 1903*; *Fraher and MacConaill, 1970*). Recent advances realized polarization microscopy also in reflection mode (*Takata et al., 2018*), and three-dimensional polarized light imaging enabled the determination of 3D fiber orientations (*Axer et al., 2011a*; *Axer et al., 2011b*; *Menzel et al., 2015*; *Zeineh et al., 2017*; *Stacho et al., 2020*; *Takemura et al., 2020*) using an advanced signal analysis (*Menzel et al., 2022*) or a tiltable specimen stage (*Schmitz et al., 2018*). However, in contrast to SLI, the techniques yield only a single fiber orientation for each measured tissue voxel, and voxels with multiple crossing fibers may yield erroneous fiber orientations (*Dohmen et al., 2015*), while retrieving the out-of-plane angle is also challenging.

To follow the course of the fiber tracts, all described sectioning techniques require accurate registration of hundreds of sections on the micrometer scale and all techniques require subsequent tractography. Tracer studies allow visualization of fiber tracts from their beginning to the end (*Lanciego et al., 2000*), but can only identify specific fiber pathways per experiment and are limited to animal brains. The only ways to follow fiber bundles in ex vivo human brains are Klinger's dissection (*Wysiadecki et al., 2019*; *Dziedzic et al., 2021*), where accuracy is limited to the macroscopic scale, or tracer injection, which is slow and impractical (*Hevner and Kinney, 1996*; *Lim et al., 1997a*; *Lim et al., 1997b*).

## Validation studies of dMRI fiber orientations

To obtain reliable connectivity maps from dMRI, a correct interpretation of the measured diffusion parameters is needed. In recent years, multiple efforts have been undertaken to enhance the interpretation of in vivo dMRI data by using post-mortem techniques as validation that provide connectional anatomy maps (*Yendiki et al., 2022*). Techniques used for validation range from histology (*Budde and Frank, 2012*; *Seehaus et al., 2015*; *Schilling et al., 2018*), serial block-face scanning electron microscopy (*Salo et al., 2018*; *Salo et al., 2021*), and microscopy of cleared tissue (*Morawski et al., 2018*; *Goubran et al., 2019*; *Leuze et al., 2021*) to polarization-sensitive optical coherence tomography (*Wang et al., 2014*; *Jones et al., 2020*; *Jones et al., 2021*) and polarized light imaging (*Caspers et al., 2015*; *Mollink et al., 2017*; *Henssen et al., 2019*; *Caspers and Axer, 2019*).

Multiple studies on simulated data and tracer studies reveal that dMRI tractography often yields false-positive fiber tracts (*Maier-Hein et al., 2017*; *Schilling et al., 2019a*; *Schilling et al., 2019b*; *Maffei et al., 2022*). Several studies indicate that dMRI orientations differ up to 20° for secondary fiber orientations and that fiber crossings at angles smaller than 60° cannot be resolved (*Schilling et al., 2016*; *Schilling et al., 2018*).

SAXS and SLI have both shown the potential to reliably resolve secondary (crossing) fiber orientations. As they provide directly structural information across extended fields of view on the same tissue sample, they might serve as validation tools for dMRI-derived fiber orientations, enabling comparisons in different anatomical regions.

## SAXS vs. SLI

Although SAXS and SLI both exploit the scattering of photons to study tissue structures, there exist fundamental differences between them. First, regarding measurement principles (*Figure 1C and D*): SAXS requires high-brilliance radiation and raster-scanning of the sample with the resolution being determined by the beam diameter and scanning step size. SLI can be performed with a simple, inexpensive setup (consisting of an LED display and camera) and provides orientation information for each camera pixel, i.e., with micrometer resolution.

While SAXS uses X-rays with ~0.1 nm wavelength, scattered at angles specific to the layer periodicity of the nerve-surrounding myelin sheath, SLI uses visible light with ~0.5 μm wavelength, interacting with the directional arrangement of myelinated axons. SLI requires several fibers on top of each other to achieve sufficient signal, whereas SAXS works already on individual (myelinated) fibers (*Inouye et al., 2014*). Also, X-ray scattering always occurs perpendicular to the nerve fibers and the pattern is center-symmetric (*Figure 2*), while SLI azimuthal profiles with an odd number of peaks cannot be interpreted without taking information from neighboring pixels into account. SAXS allows measurements of samples irrespective of sample thickness, can yield accurate fiber orientations in 3D, and can also be applied tomographically in bulk samples (*Georgiadis et al., 2021*), although the ability to tomographically resolve crossing fibers in voxels is currently limited. SLI on the other hand typically yields much higher in-plane resolutions (here: 33 x) without the time-consuming raster-scanning and can be performed with relatively inexpensive equipment in a standard laboratory without specific safety requirements.

Despite these differences, SAXS and SLI also have much in common. They are both orientation-specific methods: they directly probe the fiber orientation, without an intermediate step of imaging the tissue structures as in standard light or electron microscopy, or using a proxy such as anisotropic water diffusivity in dMRI. This enables us to reliably determine the nerve fiber orientations also in regions with densely packed, multi-directional fibers. They also result in similar azimuthal profiles for in-plane fibers, hence the same software can be used to determine peak positions. At the same time,

both techniques can image similarly-prepared tissue sections, without any staining or labeling, and they are non-destructive, enabling sample reuse e.g., for subsequent staining.

## Comparison of SAXS and SLI fiber orientations to dMRI

The 2D fiber orientations from the highly-specific SAXS measurement corresponded very well to those from the high-resolution SLI measurement (*Figures 3–4*), demonstrating the ability of both techniques to serve as ground truth for in-plane fiber orientations in complex brain tissue structures. Registering dMRI, 3D-sSAXS, and SLI data sets of a human brain sample enabled comparisons of fiber orientations from all three methods (*Figure 5* and *Figure 5—figure supplements 1–5*). When comparing the 3D-sSAXS fiber orientations of two brain sections with the corresponding dMRI-DTI fiber orientations, we observed a very high correspondence between the primary 3D fiber orientations for each voxel: the dot product is highly skewed towards one, denoting almost perfect co-alignment (*Figure 5D and E* and *Figure 5—figure supplement 2*), similar to what had been shown previously in mouse brain (*Georgiadis et al., 2020*). The regions with a low dot product (colored in blue in *Figure 5E* and *Figure 5—figure supplement 2*) are regions with two strong crossing fiber populations (*Figure 5G*), so correspondence of primary orientation is expectedly low.

When considering in-plane fibers, and especially crossing fibers in challenging regions such as the corona radiata, the fiber orientations from dMRI and SAXS also proved to be in high agreement (*Figure 5—figure supplement 3B*).

In regions with more homogeneously distributed fibers, such as in the corpus callosum (arrows in *Figure 5G*), dMRI sometimes yielded a secondary fiber population perpendicular to the main one, albeit of small magnitude (*Figure 5F* (ii) and (iii)). X-ray scattering did not show such a crossing (*Figure 5G* right), which was also missing in the micron-resolution scattered light imaging (*Figure 5H*), both showing unidirectional fibers in these voxels (regions 1 and 2). There are multiple reasons that could explain this discrepancy: First, the presence of a spurious secondary fiber population in voxels with single fiber orientations is a known artifact, caused by overfitting the response function to diffusion data (*Guo et al., 2021*; *Baete et al., 2019*), especially in the presence of noise. This is also visible in the diffusion MRI fiber orientation distributions in *Mollink et al., 2017*, Figure 8, which are not present in polarized light imaging or histology. A possible solution would be to fine-tune the model parameters so that the percentage of false-positives decreases or to increase the threshold of secondary lobes prior to running tractography algorithms, as suggested in *Maffei et al., 2022*, leading to a better agreement regarding single fiber orientations in SAXS and dMRI. However, this approach, while increasing the specificity, might decrease the sensitivity for the cases where there exist actual but less prominent secondary fiber populations. Second, the section imaged in SAXS and SLI was 80 μm thick, but was compared to dMRI voxels with 200 μm native size. This means that a dMRI voxel might contain additional fiber orientations that were not present in the section, although this seems less probable in the corpus callosum. Third, it should be noted that the current results were obtained by using a single fiber orientation analysis method (based on the MRtrix3 *dwi2response* and *dwi2fod* commands, using the *dhollander* and *msmt_csd* algorithms, respectively). Although this method is one of the most widely used for deriving fiber orientations for subsequent tractography, other methods might yield different results. So, the current results, derived for two sections from a single specimen, should be interpreted with caution.

In the present study, only fiber orientations were compared, and not fiber dispersion/orientation distributions. This has not been implemented yet in SLI and SAXS, although it would be possible to derive this information, e.g., by a process similar to the constrained spherical deconvolution in dMRI where the signal from a homogeneous fiber bundle (fiber response function) is identified and deconvolved from the signal in other voxels to obtain the fiber orientation distribution for each voxel. Given the recent development of both methods, we expect this to be performed in the following years and enrich future studies.

## Experimental validation of out-of-plane fibers in SLI

By measuring the same tissue samples with SLI and 3D-sSAXS (and dMRI for the human sample), we were able to provide experimental validation of the predicted decrease in SLI peak distance with increasing fiber inclination. However, it also became apparent that the quantification of fiber inclination based on SLI peak distance alone is challenging: while regions with steep fibers (inclinations

>70°) can be clearly identified by a high degree of scattering and small peak distances (<90°), the moderate decrease in peak distance for fibers with up to 60° inclination together with the large distribution of measured values (*Figure 6G*) makes a clear assignment between peak distance and inclination currently impossible. Our study suggests that SLI also has limitations when it comes to regions with inclined crossing fibers (*Figure 6E–F and G* on the right). To improve the interpretation, more advanced algorithms are needed. Machine learning models, trained on simulated data sets, could help to improve the interpretation of measured scattering patterns from SLI and yield more reliable estimates (suggested by *Vaca et al., 2022*).

## Quality of cross-validation

We compared results from three different imaging techniques (SLI, SAXS, dMRI) which all have different signal-generating mechanisms and resolutions. The different resolutions should be taken into account when interpreting the comparative studies: To investigate the relationship between SLI peak distance and fiber inclination, we used dMRI/SAXS images with at least 50 times lower in-plane resolution as reference (*Figure 6*). This is sufficient to validate the theoretical predictions, but insufficient to validate individual pixel values. To validate crossing fiber orientations from SAXS, we used SLI images with 30 times higher in-plane resolution, leading to a distribution of angular differences (depending on the region), but the overall narrow distribution around zero is evidence for a good overall correspondence (*Figure 4*). Finally, when comparing fiber orientations in SAXS and SLI with dMRI (*Figure 5*), it should be taken into account that dMRI voxels (with 200 µm size) contain more than double fiber layers than the corresponding SAXS or SLI voxels (with 80 µm section thickness), so that dMRI voxels might include additional fiber populations not present in our single section SAXS or SLI data. On the other hand, fiber orientations that occur both in SAXS/SLI and dMRI voxels – like the out-of-plane fiber orientations from SAXS and dMRI (e.g. *Figure 6B–C*) – can be considered as reliable, given the substantially different contrast-generating mechanisms. As SAXS and dMRI show a good correspondence for out-of-plane fiber orientations, dMRI could be used in the future to complement SLI measurements (if higher-resolution 3D-sSAXS measurements are not feasible due to measurement time), and improve the interpretation of through-plane fiber orientations.

To achieve a high quality of cross-validation, the accuracy of the image registration is crucial. The cutting process and remounting of the tissue sections might introduce non-linear distortions, requiring non-linear transformations to precisely align the images to the respective reference frame. In our case, the relatively thick microscopy sections (60 µm for vervet, 80 µm for human tissue) were not prone to significant non-linear deformations: linear transformations (scaling, rotation, translation) turned out to be sufficient to align the different image modalities. As can be seen in *Figure 4—figure supplement 1*, the registered images (middle) appear to be well aligned with the corresponding reference images (top); the boundaries of white and gray matter generally seem to coincide. Only the fornix of the vervet brain section moved during the remounting of the sample (see arrows), and was, therefore, evaluated separately, as described in Results and Methods. There might still be individual voxels that were not sufficiently well aligned, especially when comparing sections (SLI/SAXS) to volumetric measurements (dMRI). However, this would only increase angular differences between fiber orientations. The inter-method difference in our results can, therefore, be considered as an upper bound.

Another important aspect to note is that our cross-validation study focused on in-plane (single or crossing) and through-plane (single) fiber populations. At this point, we cannot make a confident assessment about highly inclined, crossing fibers. For pixel-wise comparisons as in *Figures 4 and 6*, only image pixels in which both techniques yielded the same number of fiber orientations were considered. To ensure that the determined in-plane orientations are reliable, they were only computed if the azimuthal profiles showed one/two dominant peaks (indicative of a single fiber population) or two dominant peak pairs (indicative of two in-plane crossing fiber populations), see Methods. Regions with inclined crossing or highly inclined fibers yield rather flat azimuthal profiles with multiple small peaks (*Figure 2*(iv)), so that fiber orientations cannot be compared reliably. As a result, such pixels, e.g., at the upper part of the cingulum, were excluded from the pixel-wise comparison (see *Figure 4A–C*). In addition, in-plane orientations calculated in regions with highly inclined fibers are naturally less reliable (the projection onto the plane is much smaller) and were, therefore, at this point excluded from the in-plane analysis.

## Towards a combination of SLI, SAXS, and dMRI

Our study mainly focused on the comparison and cross-validation of the different techniques. While enabling combined measurements of SLI, SAXS, and dMRI on the same tissue samples, we did not combine measurement results, e.g., in the form of combined fiber orientation maps that integrate results from all three techniques. As all techniques have their individual strengths and weaknesses, and complement each other, a combined analysis of SLI/SAXS/dMRI measurement results would greatly enhance the accuracy and reliability of brain connectivity maps. However, before this becomes feasible, several challenges need to be faced: While dMRI provides volumetric information on thick tissue samples, SLI can only be performed on thin tissue sections. SAXS tensor tomography yields fiber orientations in larger tissue volumes, but the reconstruction of several crossing fibers per tissue voxel has so far only been demonstrated in tissue sections (*Georgiadis et al., 2023*). Robust tensor tomographic reconstruction of crossing fibers or providing quantitative out-of-plane angles via SLI would allow us to fully combine the results; both developments are currently being pursued by the community. Another challenge would be to combine the fiber orientation vectors in each voxel, taking the different resolutions and accuracies of the techniques into account, for instance by fostering through-plane fiber orientations of SLI/SAXS by dMRI and verifying in-plane directions of dMRI by SLI or SAXS before using the data for tractography. We believe that tackling the issue of ground truth in fiber orientation imaging is crucial, and requires synergies between the different methods. Hence, we imagine that – after combining the methods on multiple different samples, e.g., in the way demonstrated in this study – one can identify the data portions of the highest confidence in each method and use machine learning algorithms to improve the outputs of the other methods. Eventually, this could also improve dMRI and in-vivo applications.

## Conclusion

Disentangling the highly complex nerve fiber architecture of the brain requires a combination of dedicated, multi-scale imaging techniques. We here show measurements of scattered visual light and X-ray scattering (SLI and SAXS) on the same brain tissue sample, with a high agreement between the two methods. The high-resolution properties of the former and the high-specificity of the latter can both lead to reliable detailed fiber orientation maps. The unique cross-validation of SLI, SAXS, and diffusion MRI on the same tissue sample revealed high agreement between the methods. Furthermore, it allowed the experimental validation of out-of-plane fiber orientations in SLI, paving the way for a more detailed reconstruction of 3D nerve fiber pathways in the brain. Due to the simple setup of SLI, any SAXS measurement of a tissue section can easily be complemented by a corresponding SLI measurement, which could also be mapped to prior diffusion MRI outputs. Such an approach could eventually enhance the reconstruction of nerve fiber pathways in the brain, especially in regions with complex fiber crossings.

## Materials and methods

### Vervet brain sample preparation

The vervet monkey brain was obtained from a healthy 2.4-year-old adult male in accordance with the Wake Forest Institutional Animal Care and Use Committee (IACUC #A11-219). Euthanasia procedures conformed to the AVMA Guidelines for the Euthanasia of Animals. All animal procedures were in accordance with the National Institutes of Health guidelines for the use and care of laboratory animals and in compliance with the ARRIVE guidelines. The brain was perfusion-fixed with 4% paraformaldehyde, removed from the skull within 24 hr after death, immersed in 4% paraformaldehyde for several weeks, cryo-protected in 20% glycerin and 2% dimethyl sulfoxide, deeply frozen, and coronally cut from the front to the back into 60 μm-thick sections using a cryostat microtome (*Polycut CM 3500*, Leica Microsystems, Germany). The brain sections were mounted on glass slides, embedded in 20% glycerin, and cover-slipped. Two sections from the middle (no. 511 and 501) were selected for further evaluation (see *Figure 2A* and *Figure 3—figure supplement 1A*). A region from the right hemisphere (16.4 × 10.9 mm²) – containing part of the corona radiata, corpus callosum, cingulum, and fornix – was measured with SLI several months afterward (*Figure 2B* and *Figure 3—figure supplement 1B*). For 3D-sSAXS, the brain sections were removed from the glass slides, re-immersed in phosphate-buffered solution (PBS) for two weeks, placed in-between two 170 μm-thick (#1.5) cover slips, sealed, and

measured in a comparable region (19.0 × 10.9 mm², *Figure 2C* and *Figure 3—figure supplement 1C*).

## Human brain sample preparation

The human brain (66-year-old female with no known neurological disorders) was obtained from the Stanford ADRC Biobank, which follows procedures of the Stanford Medicine IRB-approved protocol #33727, including a written informed brain donation consent of the subject or their next of kin or legal representative. The brain was removed from the skull within 24 hr, immersion-fixed for 19 days in 4% formaldehyde (10% neutral buffered formalin), coronally cut into 1 cm-thick slabs, and stored in PBS for five years. From the left hemisphere, a 3.5 × 3.5 × 1 cm³ specimen – containing part of the corona radiata, corpus callosum, and cingulum – was excised (*Figure 5A*). For dMRI, the specimen was degassed and scanned in fomblin. Five weeks later, the anterior and posterior part of the tissue was cut with a vibratome (VT1000S, Leica Microsystems, Germany) into 80 µm-thick sections. Two sections (no. 18 from the posterior side and no. 20 from the anterior side) were selected for further evaluation. For 3D-sSAXS, the brain sections were placed in-between two 150 µm-thick (#1) cover slips and measured in a center region of 28.0 × 18.9 mm² for section no. 18 (red dashed rectangle in *Figure 5B*) and 28.0 × 20.1 mm² for no. 20. For SLI, the brain sections were removed from in-between the cover slips, mounted on glass slides with 20% glycerin, cover-slipped, and measured ten weeks afterward in a region of 16.4 × 10.9 mm² containing corpus callosum and cingulum (cf. blue dotted rectangle in *Figure 5B*).

## SLI

The SLI measurements (*Figure 1D*) were performed using an LED display (*Absen Polaris 3.9pro In/Outdoor LED Cabinet*, Shenzen Absen Optoelectronic Co., Ltd., China) with 128 × 128 individually controllable RGB-LEDs with a pixel pitch of 3.9 mm and a sustained brightness of 5000 cd/m² as a light source. The images were recorded with a CCD camera (*BASLER acA5472-17uc*, Basler AG, Germany) with 5472 × 3648 pixels and an objective lens (*Rodenstock Apo-Rodagon-D120*, Rodenstock GmbH, Germany) with 120 mm focal length and 24.3 cm full working distance, yielding an in-plane resolution of 3.0 µm/px and a field of view of 16.4 × 10.9 mm². The distance between the light source and the sample was set to approximately 16 cm, and the distance between the sample and the camera to approximately 50 cm. While the in-plane resolution is determined by the magnification of the camera lens and the sensor pixel size, the through-plane resolution is limited by the thickness of the brain tissue section (60 µm for vervet and 80 µm for humans).

The SLI measurements were performed in two different ways: To generate the scattering patterns (upper *Figure 2C*), a time- and data-consuming scatterometry measurement was performed in which the sample was illuminated from 6400 different angles, as described below. This was necessary to achieve sufficiently resolved scattering patterns for a visual comparison with SAXS scattering patterns. All other results were obtained from more time- and data-efficient angular measurements in which the sample was illuminated from 24 different angles around a circle, and the fiber orientations were derived from the peak positions in the resulting azimuthal profiles.

The SLI scatterometry measurement was performed as described in *Menzel et al., 2021b*: A square of 2 × 2 illuminated RGB-LEDs (white light) was moved over the LED display in 1-LED steps for a square grid of 80 × 80 different positions, and an image was taken for every position of the square with an exposure time of 1 second. For each position of an illuminating square of LEDs, four shots were recorded and averaged to reduce noise. In the end, for each point of the sample, a scattering pattern with 80 × 80 pixels was assembled (*Figure 1D* on the right): The upper left pixel in the scattering pattern shows the intensity of the selected point in the image that was recorded when illuminating the sample from the upper left corner of the display, and so on. The azimuthal profiles in upper *Figure 2C* were generated by integrating the values of each 1° segment of the scattering pattern from the center (point of maximum intensity) to the outer border of the pattern and plotting the resulting value $I(\varphi)$ against the respective azimuthal angle ($\varphi$=0°, 1°, … 359°).

The angular SLI measurements (used to generate the SLI parameter maps in *Figures 3–6*, *Figure 2—figure supplement 1* and *Figure 3—figure supplement 1*) were performed as described in *Menzel et al., 2021a*: A rectangle of illuminated green LEDs (ca. 2.4 × 4 cm²) was moved along a circle with a fixed polar angle of illumination ($\theta$=45°) and steps of $\Delta\varphi$=15°. For every position of the rectangle

($\varphi$=0°, 15°, … 345°), an image was taken with an exposure time of 0.5 seconds. The resulting series of 24 images (containing azimuthal profiles, i.e. intensity values for each measured azimuthal angle $\varphi$ for each image pixel) was processed with the software *SLIX* (Scattered Light Imaging ToolboX) v2.4.0 (https://github.com/3d-pli/SLIX) to generate the orientational parameter maps, as described below.

## 3D-sSAXS

3D-sSAXS (*Georgiadis et al., 2015*; *Georgiadis et al., 2020*) was performed at beamline 4–2 of the Stanford Synchrotron Radiation Lightsource, SLAC National Accelerator Laboratory, with a beam of photon energy $E_{photon}$ = 15 keV. The vervet brain sections were measured (*Figure 1C*) with a beam diameter of 100 µm, an exposure time of 0.7 seconds, rotation angles $\theta$ = [0°, +/−15°,..., +/−60°], and a field of view of 19.0 × 10.9 mm² at 100 µm x- and y-steps. The human brain sections were measured with a beam diameter of 150 µm, an exposure time of 0.4 seconds, rotation angles $\theta$ = [0°, +/−10°, …, +/−70°], and a field of view of 28.0 × 18.9 mm² (posterior section no. 18) and 28.0 × 20.1 mm² (anterior section no. 20) at 150 µm x- and y-steps. While the in-plane resolution is determined by beam diameter and step size, the through-plane resolution is determined by the thickness of the tissue section (60 µm for vervet and 80 µm for humans).

To compute the in-plane fiber orientations (shown in *Figures 3–5*, *Figure 3—figure supplement 1* and *Figure 5—figure supplement 3*), azimuthal profiles were generated for each scattering pattern of the $\theta$=0°-measurement (cf. lower *Figure 2C*), and analyzed by the same SLIX software, as described below. To generate the azimuthal profiles, the scattering patterns were divided into $\Delta\varphi$=5°-segments, the intensity values were summed for each segment, and the resulting values were plotted against the corresponding average $\varphi$-value. The known center-symmetry of the SAXS scattering patterns was exploited to account for missing parts due to detector electronics.

The out-of-plane fiber inclination angles (*Figures 5C and 6A–B*) were computed by analyzing the scattering patterns obtained from 3D-sSAXS measurements at different sample rotation angles, as described in *Georgiadis et al., 2020*.

## Diffusion MRI

The dMRI measurement on the human brain specimen was performed on a Bruker 11.7T scanner, using a 12-segment spin-echo echo planar imaging (SE-EPI) sequence at 200 µm isotropic voxels, repetition time TR = 400 ms, echo time TE = 40 ms, diffusion separation time $\delta$ = 7 ms, diffusion time $\Delta$ = 40 ms, field of view FOV = 40 × 36 × 21 mm³, at 200 diffusion-weighted $q$-space points (20@b = 1 ms/µm², 40@b = 2 ms/µm², 60@b = 5 ms/µm², 80@b = 10 ms/µm²) and 20@b = 0 ms/µm². First, data were denoised and corrected for Gibbs artifacts (*Ades-Aron et al., 2018*; *Veraart et al., 2016*). Then, volumes were b-value-averaged, and registered to the initial b0 volume using *FSL FLIRT* (https://fsl.fmrib.ox.ac.uk/fsl/fslwiki/FLIRT; *Jenkinson et al., 2002*) with mutual information as a cost function and a spline interpolation. After registration to the SLI and SAXS (see corresponding Methods section 'Image registration and pixel-wise comparison'), fiber responses and orientation distributions were computed using the *dwi2response* and *dwi2fod* functions in MRtrix3 (https://www.mrtrix.org/) –employing *dhollander* and *msmt_csd* (multi-tissue, multi-shell constrained spherical deconvolution with spherical harmonics up to 8th degree) algorithms, respectively– and visualized in *mrview*. The dMRI-derived output fiber orientation distributions for each voxel were sampled at the plane of the vibratome section in 5°-steps using MRtrix3's *sh2amp* command, which was then used as input to the SLIX software package for computing in-plane fiber orientations including crossings. For main fiber orientations, diffusion tensor imaging (DTI) processing was performed using FSL's *DTIFIT* function (https://fsl.fmrib.ox.ac.uk/fsl/fslwiki/FDT/UserGuide#DTIFIT). For the dMRI maps (*Figure 5—figure supplement 1*), the DESIGNER pipeline (https://github.com/NYU-DiffusionMRI/DESIGNER; *Ades-Aron et al., 2018*) was used to compute diffusivity, kurtosis, and white matter tract integrity parameters (*Fieremans et al., 2011*).

## Generation of orientational parameter maps

The azimuthal profiles from angular SLI, 3D-sSAXS, and dMRI were processed with SLIX in order to generate various parameter maps (*Figure 2—figure supplement 1*) and to determine the in-plane nerve fiber orientations. The analysis of the profiles and the SLIX software are described in *Menzel et al., 2021a* and *Reuter and Menzel, 2020* in more detail. The software determines the positions of

the peaks for each image pixel (azimuthal profile). The peak prominence (*Figure 2—figure supplement 1*, 4th row) was determined as the vertical distance between the top of the peak and the higher of the two neighboring minima. Only peaks with a prominence larger than 8% of the total signal amplitude (max – min) were considered for evaluation. The peak width (*Figure 2—figure supplement 1*, last row) was computed as the full width of the peak at a height corresponding to the peak height minus half of the peak prominence. The in-plane fiber orientation $\varphi$ (*Figures 3–5*, *Figure 3—figure supplement 1* and *Figure 5—figure supplement 3*) was computed as the mid-position between peaks that lie 180°+/−35° apart. To better analyze multiple crossing fiber orientations, the in-plane fiber orientations were visualized as colored lines and displayed on top of each other (*Figure 3* and *Figure 3—figure supplement 1*). The peak distance (*Figure 6D–E*) was computed as the distance between two peaks, for profiles with no more than two peaks (profiles with one peak yield zero peak distance). The peak distance of the SLI azimuthal profiles was compared to the through-plane fiber orientations from 3D-sSAXS and dMRI to study the relationship between decreasing peak distance in SLI and increasing fiber inclination in SAXS and dMRI.

## Image registration and pixel-wise comparison

To minimize loss of information, the pixel/voxel-wise comparisons were performed at the spacing of the highest resolution image, i.e., the lower-resolution dMRI/SAXS images were upscaled to the higher-resolution SLI images. As a result, the fiber orientation from a single 150 µm SAXS pixel, for example, was compared to 50 × 50 SLI fiber orientations (px = 3 µm). In voxels with crossing fiber orientations, it was ensured that the first and second orientation matched across modalities, i.e., the first fiber orientation of one modality was compared to both orientations of the other modality, and the one with the smaller angular difference was used for comparison. The second orientation of the first modality was then compared to the remaining orientation of the second modality.

To register 3D-sSAXS onto SLI (*Figures 4 and 6*), 9 degree of freedom linear registration (scaling, rotation, translation) was performed using FSL FLIRT (https://fsl.fmrib.ox.ac.uk/fsl/fslwiki/FLIRT), while angular information and 3D vectors were rotated accordingly. The fornix of the vervet brain section was separately registered as it was only connected to the rest of the tissue by a thin strip and had moved when re-mounting the section. For registering dMRI onto SAXS (*Figure 5* and *Figure 5—figure supplements 2–3*), first the matching plane for each human brain section was identified manually in the scanned MRI volume (different plane for each human brain section), and FSL FLIRT linear registration with 9 degrees of freedom was used for precise alignment of the 2D images. Then, the entire dMRI data set was transformed using the identified rotation and translation parameters (twice, once for each section), and the *b*-vectors were rotated correspondingly. The MRI sections corresponding to the vibratome section plane were isolated and further analyzed as explained in the 'Diffusion MRI' Methods section. A comparison of reference and registered images is shown in *Figure 4—figure supplement 1* for vervet brain section no. 511 and the posterior human brain section.

## Code and data availability

All software used for image processing is open-source and described in the Methods section (with corresponding URL links). Data analysis for *Figures 4 and 6* was performed with Fiji (https://fiji.sc/Fiji), for *Figure 5* and supplementary figures using Matlab 2021b (Mathworks, USA). All images and parameter maps used for this study are publicly available at original resolution (https://doi.org/10.5281/zenodo.7208998).

## Acknowledgements

We thank Laura Pisani from the Stanford Center for Innovation in In vivo Imaging (SCi3) and Kristin Garlund from Bruker BioSpin USA for assistance with the dMRI measurements, Roger Woods from the UCLA Brain Research Institute and Donald Born from Stanford Pathology for providing the vervet and human brain samples, respectively, and the laboratory team from the Institute of Neuroscience and Medicine (INM-1), Forschungszentrum Jülich, for the preparation of the vervet and human brain sections for the SLI measurements. This work was supported by the National Institutes of Health (NIH) under award numbers R01NS088040, P41EB017183, R01AG061120-01, R01MH092311, and 5P40OD010965, by the Helmholtz Association port-folio theme 'Supercomputing and Modeling for the Human Brain,' and by the European Union's Horizon 2020 Research and Innovation Programme

under Grant Agreement No. 945539 ('Human Brain Project' SGA3). The Stanford Synchrotron Radiation Lightsource, SLAC National Accelerator Laboratory, is supported by the U.S. Department of Energy, Office of Science, Office of Basic Energy Sciences under Contract No. DE-AC02-76SF00515. The SSRL Structural Molecular Biology Program is supported by the DOE Office of Biological and Environmental Research, and by the National Institutes of Health, National Institute of General Medical Sciences (P30GM133894). The Pilatus detector at beamline 4–2 at SSRL was funded under National Institutes of Health Grant S10OD021512. M.M. received funding from the Helmholtz Association of German Research Centers (Helmholtz Doctoral Prize 2019) and the Klaus Tschira Stiftung gGmbH (Klaus Tschira Boost Fund).

## Additional information

### Funding

| Funder | Grant reference number | Author |
| --- | --- | --- |
| Helmholtz Association | Supercomputing and Modeling for the Human Brain | Miriam Menzel |
| Helmholtz Association | Helmholtz Doctoral Prize 2019 | Miriam Menzel |
| Horizon 2020 Framework Programme | Human Brain Project SGA3 (945539) | Miriam Menzel David Gräßel |
| Klaus Tschira Stiftung | Klaus Tschira Boost Fund | Miriam Menzel |
| National Institutes of Health | R01AG061120-01 | Michael M Zeineh |

The funders had no role in study design, data collection and interpretation, or the decision to submit the work for publication.

### Author contributions

Miriam Menzel, Conceptualization, Formal analysis, Supervision, Funding acquisition, Investigation, Visualization, Methodology, Writing – original draft, Writing – review and editing, SLI analysis/evaluation; David Gräßel, Investigation, Writing – review and editing, SLI measurements; Ivan Rajkovic, Investigation, Writing – review and editing, SAXS measurement assistance; Michael M Zeineh, Conceptualization, Resources, Formal analysis, Supervision, Funding acquisition, Writing – review and editing; Marios Georgiadis, Conceptualization, Formal analysis, Supervision, Funding acquisition, Investigation, Visualization, Methodology, Writing – original draft, Writing – review and editing, SAXS/dMRI measurements/evaluation

### Author ORCIDs

Miriam Menzel http://orcid.org/0000-0002-6042-7490
David Gräßel http://orcid.org/0000-0003-3228-8048
Marios Georgiadis http://orcid.org/0000-0003-0733-4559

### Ethics

Human subjects: The human brain tissue was obtained from the Stanford ADRC Biobank, which follows procedures of the Stanford Medicine IRB-approved protocol #33727, including a written informed brain donation consent of the subject or their next of kin or legal representative.
The vervet monkey brain tissue was obtained in accordance with the Wake Forest Institutional Animal Care and Use Committee (IACUC #A11-219). Euthanasia procedures conformed to the AVMA Guidelines for the Euthanasia of Animals. All animal procedures were in accordance with the National Institutes of Health guidelines for the use and care of laboratory animals and in compliance with the ARRIVE guidelines.

### Decision letter and Author response

Decision letter https://doi.org/10.7554/eLife.84024.sa1

Author response https://doi.org/10.7554/eLife.84024.sa2

## Additional files

### Supplementary files
• MDAR checklist

### Data availability

All data generated and analyzed in this study are included in the manuscript and supporting figures. The corresponding high-resolution images and parameter maps have been deposited in Zenodo under https://doi.org/10.5281/zenodo.7208998.

The following dataset was generated:

| Author(s) | Year | Dataset title | Dataset URL | Database and Identifier |
|---|---|---|---|---|
| Menzel M, Gräßel D, Rajkovic I, Zeineh M, Georgiadis M | 2022 | Dataset: Using light and X-ray scattering to untangle complex neuronal orientations and validate diffusion MRI | https://zenodo.org/record/7208998 | Zenodo, 10.5281/zenodo.7208998 |

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
