## [Editor Report]

This paper presents a valuable cross-validation study of mesoscopic measurements of axonal orientations from three different modalities – small-angle X-ray scattering, scattered light imaging, and diffusion MRI – and is of interest to researchers who want to apply these methods to white matter imaging. The authors show convincing similarities and differences in fiber orientations from all three methods over partial ex vivo brain samples, though the validation of diffusion MRI could be strengthened by considering other fiber reconstruction methods, as only a single diffusion method is investigated.

---

## [Decision Letter]

**Decision letter after peer review:**

Thank you for submitting your article "Using light and X-ray scattering to untangle complex neuronal orientations and validate diffusion MRI" for consideration by *eLife*. Your article has been reviewed by 2 peer reviewers, one of whom is a member of our Board of Reviewing Editors, and the evaluation has been overseen by Timothy Behrens as the Senior Editor. The following individual involved in the review of your submission has agreed to reveal their identity: Anastasia Yendiki (Reviewer #2).

Essential revisions:

1) One primary concern is regarding the generalisability of the conclusions with respect to the diffusion MRI analysis. First, diffusion MRI provides many different approaches to estimating fibre orientations within the voxel, each of which often has user-defined options that can affect the resulting fibre orientation distribution. Here only a single method is presented. Second, the presence of secondary peaks of small magnitude in single fibre voxels such as a corpus callosum is a known artefact, where subsequent downstream analyses often ignore peaks below some magnitude threshold. A better agreement between SAXS and diffusion MRI may be found if the diffusion FODs were thresholded prior to comparison. Third, the diffusion voxels are larger than the microscopy sections, meaning that the microscopy may not include fibre populations existing in the MRI. Fourth, the analysis considers two postmortem brain samples covering only a small tissue area.

Though the results on diffusion MRI may reflect the data and analysis presented, the conclusions lack generalisability as the diffusion MRI estimates of FOD are very methods dependent (with only one tested here), and potentially lack impact as these additional peaks are often ignored in downstream analyses. These points were somewhat noted in the discussion but do strongly limit the interest of these results to the diffusion MRI community. Given these limitations, the authors should refrain from making unsupported generalised claims about all diffusion MRI analyses, and considerable changes are required to ensure this key concern is properly addressed. This includes but is not limited to textual edits throughout the manuscript and the removal of titles such as "Diffusion MRI tends to overestimate fiber orientations in the human brain" and similar phrases. The title "Identification of false-positive fiber tracts in dMRI", should also be reconsidered given that analysis to identify tracts (e.g. tractography) rather than fibre populations was not performed.

2) The comparison of fibre orientations at the pixel/voxel-wise level is heavily reliant on the accuracy of the data co-registration, including rotation of the fibre orientations. Here only affine registration was used, though the microscopy processing may result in non-linear deformations. Please can the authors provide an evaluation of the quality of the coregistration and a discussion into how this may affect results?

3) Some of the methods were unclear and require additional clarification (please see comments below). In particular, it was unclear a) at what resolution the SLI/SAXS comparisons were made and how data were up/down-sampled to make these comparisons, b) the difference between SLI scatterometry and angular SLI, c) how through plane orientations are computed from SLI, and d) how the correspondence of primary and secondary fibre populations was ensured across modalities.

4) The presentation of the study as a framework for combining SLI/SAXS/dMRI data, rather than a cross-validation study, caused some confusion. There are no clear prescriptions for how researchers interested in the neuroanatomical investigation would combine information from SLI and SAXS, or use the two methods in tandem, rather the information from one modality is used to corroborate/validate that from the other. Further, only SAXS is really compared to the diffusion MRI, presumably due to SLI having unreliable through-plane information. The authors will need to re-consider wording throughout the manuscript (see phrases such as "The combination of scattered light and X-ray imaging …[enable] detailed investigations of complex tract architecture in the animal and human brain", "The high-resolution properties of the former combined with the high-specificity of the latter enables the detailed reconstruction of multiple nerve fiber orientations for each image pixel, which can provide providing unprecedented insights into brain circuitry." or "The presented framework can … deliver more accurate brain connectivity maps of the animal and human brain." for example), or else additional information and example analyses should be provided on how these modalities can be combined for neuroanatomical investigation, particularly in regions where the information from the different modalities does not agree.

5) This may be an issue with the paper submission, but unfortunately the PDF figures were low resolution making the fibre orientations difficult to see. Higher-resolution images would be required in print.

This study presents a valuable comparison of fibre orientation estimates from three different modalities: diffusion MRI, scattered light imaging, and x-ray scattering. The comparison is interesting as each modality is sensitive to different aspects of tissue microstructure – water anisotropy, micron-scale structural coherence, and myelin lamella respectively. Where scattered light and x-ray imaging can be only applied ex vivo, diffusion MRI has in vivo applications but suffers from being an indirect estimate of the microstructure of interest. By acquiring all modalities in both a vervet monkey and human brain sample, the authors provide quantitative, pixel/voxel-wise comparisons of fibre orientation estimates within the same tissue samples. The authors show convincing agreement in fibre orientations from all three methods, giving confidence in the fidelity of the methods for neuroanatomical investigations. Differences are also observed: SLI is shown to have less reliable estimates of fibre inclination, and the CSD analysis presented overestimates the number of crossing fibre populations when compared to the microscopy methods, particularly in single fibre regions such as the corpus callosum, a known artefact in some diffusion analyses.

In the current PDF, it is very difficult to see fibre orientations in figures due to low resolution, limiting the reader's ability to assess the results. Higher-resolution images would provide more information and easier comparisons.

The methods are generally clear though some additional information is needed: 1) to specify the resolution that the orientations are compared in each figure and how data was up-/down-sampled for these comparisons respectively. For example, each SAXS pixel contains many SLI pixels. It is currently unclear whether the mean SLI orientation from a neighbourhood is equivalent to the SLI compared, or whether a comparison was made for each SLI pixel. Similarly, for the dMRI-microscopy comparisons. 2) I also could not follow why two SLI methods are presented in the methods: SLI scatterometry relating to Figure 2, and angular SLI relating to all other results. Further clarification is needed. 3) Since the quality of the data co-registration can strongly impact pixel/voxel-wise comparisons, quantification of the registration accuracy or overlays demonstrating the quality of the co-registration would be valuable.

A primary weakness of the work as a diffusion MRI validation study is that though diffusion MRI supports many different models to extract fibre orientations with different outputs, here only a single model is compared to the microscopy data, which may affect the generalisability of the results. Further, it only compares the primary orientations from the diffusion MRI and does not consider each fibre population's magnitude (density of fibres) or the orientation dispersion, both of which can influence downstream analyses.

The paper could be strengthened by a more detailed discussion on the differences between the imaging modalities – e.g. in terms of imaging resolution, signal-generating mechanisms, and sensitivity to specific aspects of the tissue microstructure – and how these differences may limit their application to specific neuroanatomical investigations, or ability to validate one another. For example, the microscopy sections are 80 microns thick whilst the diffusion voxel is 200 microns. I expect this could contribute to the difference in the number of fibre populations per voxel.

The hypothesis that dMRI signal contributions from extra-axonal water result in additional fibre populations could be investigated by running CSD on both low and high-b-value data (for example using the openly available MGH dataset, Fan 2016) where fewer secondary fibre populations should be observed at high b-value.

Thanks to the authors for providing links to the code and data relevant to the manuscript.

It would be very interesting to have the authors comment on the ability of SLI/SAXS to validate dMRI estimates of fibre density of dispersion or demonstrate a comparison of these.

Some suggestions on the manuscript:

– Abstract: "we find a reduction of peak distance with increasing out-of-plane fiber angles" – Without additional context from the paper, I found this statement unclear. Some further explanation or rephrasing would help.

– Line 58 – Please can you give the resolutions of SLI/SAXS

– Line 123 "Figure 4A shows very small angular differences that appear to be uniformly distributed" – I'm not sure what is meant here the distribution doesn't appear to be shown.

– Figure 4 – please can you clarify why some regions do not appear to be included in the analysis? i.e. the top right-hand side of the slide is missing in A/B and parts of the cingulum appear to be missing in part C. What is the rationale for this? By excluding many pixels from the cingulum in the histogram in 4d, would this not lead to an underestimation of the angular difference?

– Figure 4a – I would recommend using a colour bar where colours have a unique meaning. White or light red currently can mean more than one thing in the image.

– Figure 5 – The 3D colour wheel is atypical for dMRI community which usually uses blue for IS and green for AP. Though the current version is acceptable as the meaning of the colours is clearly indexed, it may be worth considering changing to avoid any confusion.

– Line 173 onwards – Please can you provide the maps of single/crossing fibre voxels for both dMRI and SAXS? It would be interesting to see the clustering of the single/crossing fibre populations.

– Line 203 – I find it very hard to interpret or see the 3D arrows in Figure 6A

– Line 413 – "The high-resolution properties of the former combined with the high-specificity of the latter enables the detailed reconstruction of multiple nerve fiber orientations for each image pixel, which can provide providing unprecedented insights into brain circuitry." This conclusion seems out of place given the work presented, as details have not been provided on how to combine information from these modalities.

– Line 414 – Typo "provide providing"

– Line 429 – I assume the tissue was immersion fixed?

*Reviewer #2 (Recommendations for the authors):*

This work is a cross-validation of an x-ray tomography technique (SAXS) and an optical microscopy technique (SLI) for imaging axonal orientations ex vivo. These innovative methods were introduced in recent papers by the authors, who have teamed up here to compare them side-by-side on the same tissue samples for the first time. The two methods are both label-free (do not require staining) and they are quite complementary. SAXS can provide full 3D orientation measurements on intact tissue, but it operates at a mesoscopic resolution and requires access to a synchrotron. SLI can measure the orientations of multiple fascicles per voxel at a microscopic resolution and relies on more widely accessible equipment, but its accuracy suffers for fiber orientations perpendicular to the imaging plane and it requires tissue to be sectioned before it is imaged. Therefore it makes a lot of sense to explore the complementary strengths of these two techniques, and to use one to "fill in the blanks" of the other. The paper also compares the orientation measurements obtained with SAXS and SLI to those obtained with diffusion MRI. The latter provides only indirect measurements based on water diffusion, at a mesoscopic resolution somewhat lower than that of SAXS, but has the benefit of being feasible in vivo.

A limitation of this study is that conclusions on the comparison between SAXS and SLI are drawn from only 2 sections of a partial monkey brain sample and 2 sections of a partial human brain sample. Conclusions on diffusion MRI are drawn only on the 2 human sample sections. This is particularly an issue for the comparison to diffusion MRI, as the diffusion MRI voxels are wider than the section thickness, hence one cannot preclude that any orientations detected with diffusion MRI but not with SAXS and SLI come from the portion of the voxel that is missing from the corresponding SAXS/SLI section.

The stated aim of the paper is to provide a framework for combining the complementary benefits of SAXS and SLI, rather than simply presenting the results of a cross-validation study. This is a significant and ambitious aim. However, in order for this to serve as a framework, there would have to be clear prescriptions for how researchers interested in obtaining ground-truth measurements of axonal orientations would do so by using these two methods in tandem. This is not adequately developed in the paper in its present form. For example, the results show reasonable agreement between SAXS and SLI orientations when fibers lie within the SLI imaging plane and decreasing agreement for fibers with increasing through-plane inclination. How would the two methods be combined in voxels where they disagree? Would one use SLI orientations in voxels with fewer through-plane fibers and SAXS orientations in voxels with more through-plane fibers? How would voxels be assigned to each category? How would the orientation vectors from the two modalities be composed and how would the resolution difference between the two be handled? When the through-plane measurement of SLI is unreliable, is its in-plane measurement still reliable? That is if there were one mainly in-plane and one mainly through-plane fiber population, would the orientation of the former still be measured correctly by SLI? There is also considerable agreement reported here between through-plane orientations obtained with SAXS and diffusion MRI. Would this mean that diffusion MRI itself could be used to supplement SLI with through-plane orientations? Any clear set of prescriptions along these lines would represent a framework for imaging orientations by combining modalities. This, however, would require detailed steps for how to perform the combination and use the multi- vs. uni-modal framework to reconstruct connectional anatomy.

A key advantage of SAXS is that it can be performed on intact samples, i.e., before any nonlinear distortions of the tissue are introduced by sectioning. Thus it can provide an undistorted reference, with contrast on axonal orientations that would be absent in, say, a structural MRI of comparable resolution. This contrast could be used to drive registration of the distorted SLI sections to an undistorted SAXS volume, and therefore is a key way in which the two techniques can complement each other. Here, however, this is not explored, as SAXS is performed after sectioning. It is not clear if this is the authors' prescription for how a combined SAXS/SLI framework would be implemented, or if it was done specifically for this study. First, it would seem that SAXS on the intact sample would be lower maintenance, requiring less setup time and hence potentially less overall beamtime than performing SAXS on each section separately. This would make it more practical for routine deployment beyond a few sections. Second, because the SAXS data are now nonlinearly distorted, they cannot be affinely aligned to the MRI volumes. While, in principle, performing both SAXS and SLI on the sections may facilitate the comparison between the two, having to unmount, rehydrate, and remount the sections in between may negate this advantage, as now there is no guarantee that SAXS and SLI can be affinely registered to each other. Here all these registration steps are performed affinely, so it is unclear to which extent the computed errors between modalities are characterizing the inherent limitations of the respective contrasts, or limitations of the registration technique. Some of the alignment is performed manually, for example, specific regions of the images are realigned by hand, and the slice of the diffusion MRI volume that is aligned to the SAXS/SLI sections is chosen by hand. Again, for this to serve as a framework that can be deployed on whole samples, there would have to be clear prescriptions for how to perform these steps robustly, how to ensure that the MRI can be acquired in a coordinate frame parallel to the sections, etc.

Finally, the paper puts forth a general conclusion that diffusion MRI overestimates the number of fiber populations per voxel, on the basis of small ODF peaks appearing perpendicular to the main ODF peaks. Of all conclusions in the paper, this is the least convincingly supported by evidence. First, these small perpendicular peaks are a known artifact, which would be typically eliminated by ignoring ODF peaks below a certain amplitude, a common practice in diffusion tractography algorithms. The authors refrain from using an amplitude threshold, with the rationale that it may also remove true diffusion orientations. However, they apply a threshold when they detect SLI peaks (a rather stringent 8% of the maximum). Second, the explanation that these artifactual peaks may appear due to vessel walls is not convincing. Vasculature is sparse. A single vessel wall will not impact the diffusion signal in the same way as a bundle of parallel axons. In an axon bundle, water molecule displacements are restricted in all directions except parallel to the axons. A single vessel wall in a voxel will not have the same effect on displacements (which are much smaller than the size of the voxel). From Figure 5, it looks like there would be at most 1-2 of these vessels in a diffusion MRI voxel, and they would not be in all voxels. This cannot explain the widespread appearance of these small artifactual peaks. Third, many ODF reconstruction methods have parameters that can be adjusted to make these artifactual peaks more or less prominent. The default parameters may be optimal for in vivo but not ex vivo data, due to the effects of fixation. In light of these concerns, I would caution against making such a general statement about all diffusion MRI in the human brain, especially on the basis of a single diffusion reconstruction method applied to a single location in one brain.

– The term "neuronal orientations" (or "trajectories") is used throughout but "axonal orientations" would be more suitable, as neurons are not imaged here.

– "… the signal is affected by all brain structures, not axons". It is true that the diffusion signal is affected by all tissue components, but that is not the main issue. What we use to extract orientations is not the signal itself, but the anisotropy (orientational dependence) of this signal. Not all tissue components contribute to this anisotropy the same way that myelin sheaths and axon membranes do, even if they contribute to the signal itself.

– There are several mentions of false positives as the problem with diffusion tractography methods. However no method has a fixed false positive rate, all methods have thresholds that can be adjusted to make their false positive rate as low or as high as one wants. It's the trade-off between the false positive rate and true positive rate that is difficult to improve, not the false positives themselves. You can always remove false positives, but the challenge is that you cannot do it without an excessive loss of true positives.

– In Methods: the in-plane resolution of the SAXS and SLI scans is provided, but what is the through-plane resolution? Is it the same as the section thickness (80 microns)?

– In Methods: there is a description of how the in-plane and through-plane orientations are computed in SAXS, but only how the in-plane orientations are computed in SLI.

– In Methods: it is unclear at what scale the angular errors are computed. Is it at the highest resolution among the modalities that are being compared, i.e., do you obtain as many values of the error as SLI-sized voxels?

– The use of the word "peak" without further qualifiers is likely to confuse the diffusion MRI audience. Here 2 peaks (in the SLI profile) = 1 fiber population, but in usual diffusion terminology 1 peak (in the diffusion/fiber ODF) = 1 fiber population.

– How are the "primary" and "secondary" orientations in a voxel determined for each modality? Are any steps taken to ensure that they match across modalities? Otherwise, you can imagine a scenario where the same fiber population ends up being the primary orientation in one modality but the secondary one in the other, and vice versa. This would overestimate the angular error between modalities. If the goal is to measure error as a difference between orientation angles, not amplitudes (as amplitudes mean different things in each modality anyway), then care must be taken to establish the correspondence of fiber populations between modalities.

– Registration with 12 degrees of freedom is affine. Linear and affine are not quite synonyms.

– It would be helpful to report the angular resolution of each technique as a function of through-plane inclination, e.g., what is the smallest angle between crossing fibers that can be accurately resolved by SLI when the fibers are in-plane or inclined?

– In Discussion: the paragraph on clearing and fluorescence microscopy should be updated to reflect the latest developments in that field. It is not "only feasible for smaller sample sizes", as clearing has now been demonstrated in samples as large as a whole slab of a human hemisphere. Also, the statement "it fails to disentangle densely packed nerve fibers" is not generally true, as this depends on the imaging resolution. The resolution of fluorescence microscopy can in principle be increased to the sub-micron level. It is true that the few studies that compared fluorescence microscopy of cleared tissue to diffusion MRI so far did not use state-of-the-art of these methods, but this is not due to an inherent limitation of the methods.

– In Discussion: while a single PS-OCT measurement gives only the in-plane orientation, the use of multiple measurements at multiple incidence angles to infer the through-plane orientation has been demonstrated in the literature.

– In Discussion: the statement that SAXS and SLI yield orientations with "a higher precision and smaller crossing angles" should be clarified. Higher/smaller than what? Which modality here is used as the reference that SAXS, SLI, and diffusion MRI are being compared to, in order to determine that the error is lower in one than the other?

---

## [Author Response]

Essential revisions:1) One primary concern is regarding the generalisability of the conclusions with respect to the diffusion MRI analysis. First, diffusion MRI provides many different approaches to estimating fibre orientations within the voxel, each of which often has user-defined options that can affect the resulting fibre orientation distribution. Here only a single method is presented. Second, the presence of secondary peaks of small magnitude in single fibre voxels such as a corpus callosum is a known artefact, where subsequent downstream analyses often ignore peaks below some magnitude threshold. A better agreement between SAXS and diffusion MRI may be found if the diffusion FODs were thresholded prior to comparison. Third, the diffusion voxels are larger than the microscopy sections, meaning that the microscopy may not include fibre populations existing in the MRI. Fourth, the analysis considers two postmortem brain samples covering only a small tissue area.

We understand this concern, and have included all these valid points in the revised version of our manuscript.

Changes: We have addressed the points mentioned above, among others in the newly named Discussion section “Comparison of SAXS and SLI fiber orientations to dMRI” and the new Discussion section “Quality of cross-validation”:

“it should be noted that the current results were obtained by using a single fiber orientation analysis method (based on the MRtrix3 dwi2response and dwi2fod commands, using the dhollander and msmt_csd algorithms, respectively). Although this method is one of the most widely used for deriving fiber orientations for subsequent tractography, other methods might yield different results. So, the current results […] should be interpreted with caution.”“the presence of a spurious secondary fiber population in voxels with single fiber orientation is a known artifact, caused by overfitting the response function to diffusion data (Guo et al., 2021; Baete et al., 2019), especially in the presence of noise. This is also visible in the diffusion MRI fiber orientations distributions in Mollink et al. (2017), Figure 8, which are not present in PLI or histology. A possible solution would be to fine-tune the model parameters so that the percentage of false-positives decreases or to increase the threshold of secondary lobes prior to running tractography algorithms, as suggested in Maffei et al. (2022), leading to a better agreement regarding single fiber orientations in SAXS and dMRI. However, this approach, while increasing the specificity, might decrease the sensitivity for the cases where there exist actual but less prominent secondary fiber populations.”“the section imaged in SAXS and SLI was 80μm thick, but was compared to dMRI voxels with 200μm native size. This means that an dMRI voxel might contain additional fiber orientations that were not present in the section, although this seems less probable in the corpus callosum.” And: “when comparing crossing fibers in SAXS and SLI with dMRI (Figure 5), it should be taken into account that dMRI voxels (with 200µm size) contain more fiber layers than the corresponding SAXS or SLI voxels (with 80µm section thickness), so that dMRI voxels might include additional fiber populations not present in SAXS or SLI data”“the current results, derived from two sections of a single specimen, should be interpreted with caution.”

Though the results on diffusion MRI may reflect the data and analysis presented, the conclusions lack generalisability as the diffusion MRI estimates of FOD are very methods dependent (with only one tested here), and potentially lack impact as these additional peaks are often ignored in downstream analyses. These points were somewhat noted in the discussion but do strongly limit the interest of these results to the diffusion MRI community. Given these limitations, the authors should refrain from making unsupported generalised claims about all diffusion MRI analyses, and considerable changes are required to ensure this key concern is properly addressed. This includes but is not limited to textual edits throughout the manuscript and the removal of titles such as "Diffusion MRI tends to overestimate fiber orientations in the human brain" and similar phrases. The title "Identification of false-positive fiber tracts in dMRI", should also be reconsidered given that analysis to identify tracts (e.g. tractography) rather than fibre populations was not performed.

We agree that our findings should not be understood as generalizable for all dMRI analyses and have considerably changed our manuscript to properly address this key concern.

As suggested, we have removed all phrases that could be understood as generalized claims for MRI (including any reference to fiber orientations overestimation), and changed the section title “Diffusion MRI tends to overestimate fiber orientations in the human brain” to “Comparison to diffusion MRI shows high agreement and small discrepancies”. We have also removed any mention of “false-positive fiber tracts” in the Introduction and Conclusion, and changed the section title “Identification of false-positive fiber tracts in dMRI” to “Comparison of SAXS and SLI fiber orientations to dMRI”. Furthermore, we have changed the wording throughout the manuscript (see also above examples) to discuss the non-generalizability of our results.

2) The comparison of fibre orientations at the pixel/voxel-wise level is heavily reliant on the accuracy of the data co-registration, including rotation of the fibre orientations. Here only affine registration was used, though the microscopy processing may result in non-linear deformations. Please can the authors provide an evaluation of the quality of the coregistration and a discussion into how this may affect results?

It is true that the accuracy of co-registration has a decisive impact on the pixel/voxel-wise comparison of the fiber orientations. In our case, we could prevent significant non-linear deformations (possibly due to the relatively thick microscopy sections of 60μm and 80μm) and achieve very good overlap of tissue structures without using non-linear terms (with exception for the vervet fornix, see below). The fact that we found standard linear transformations (scaling, rotation, and translation) to be sufficient for achieving a fair comparison between the different modalities also demonstrates the experimental feasibility of our approach.

To help the reader evaluate the quality of the co-registration, we have generated a new figure (Figure 4 —figure supplement 1) which shows the reference and registered images next to each other, as well as an overlay of the different modalities (as suggested by Reviewer #1). For better comparison, the white/gray matter boundaries of the reference image are displayed in the registered images as red outlines.

The figure shows a good correspondence between reference and registered images. The only region that does not show a good correspondence is the fornix of the vervet brain section (see arrows in this new figure) which moved when re-mounting the section. Therefore, the fornix was evaluated separately, as stated in the manuscript.

Although reference and registered images show a good overall correspondence, there might still be voxels that are not sufficiently well aligned and lead to larger differences when comparing the fiber orientations. The results on the differences between the methods presented here can therefore be considered as upper bound. Using standard linear transformations, we could already show that the fiber orientations between SLI and SAXS correspond very well, also in regions with fiber crossings. A more accurate registration would only yield an even better correspondence of fiber orientations.

We have added the supplementary figure, Figure 4 —figure supplement 1, referred to it in the Results/Methods sections, and added the following paragraph to the new Discussion section “Quality of cross-validation”:

“To achieve a high quality of cross-validation, the accuracy of the image registration is crucial. The cutting process and remounting of the tissue sections might introduce non-linear distortions, requiring non-linear transformations to precisely align the images to the respective reference frame. In our case, the relatively thick microscopy sections (60μm for vervet, 80μm for human tissue) were not prone to significant non-linear deformations: linear transformations (scaling, rotation, translation) turned out to be sufficient to align the different image modalities. As can be seen in Figure 4—figure supplement 1, the registered images (middle) appear to be well aligned with the corresponding reference images (top); the boundaries of white and gray matter generally seem to coincide. Only the fornix of the vervet brain section moved during the remounting of the sample (see arrows) and was therefore evaluated separately, as described in Results and Methods. There might still be individual voxels that were not sufficiently well aligned, especially when comparing sections (SLI/SAXS) to volumetric measurements (dMRI). However, this would only increase angular differences between fiber orientations. The inter-method difference in our results can therefore be considered as an upper bound.”

3) Some of the methods were unclear and require additional clarification (please see comments below). In particular, it was unclear a) at what resolution the SLI/SAXS comparisons were made and how data were up/down-sampled to make these comparisons.

To minimize loss of information, the pixel-wise comparisons (as presented in Figures 4 and 6) were made at the spacing of the highest resolution image: Before comparing dMRI/SAXS to SLI, the lower-resolution dMRI/SAXS images were upscaled to match the higher-resolution SLI images. This means, for example, that one SAXS fiber orientation (px=150µm) was compared to 50x50 SLI fiber orientations (px=3µm).

We have added the paragraph below to the newly named Methods section (“Image registration and pixel-wise comparison”), and specified in each figure caption at which resolution the images were compared (suggested by Reviewer #1).

“To minimize loss of information, the pixel/voxel-wise comparisons were performed at the spacing of the highest resolution image, i.e. the lower-resolution dMRI/SAXS images were upscaled to the higher-resolution SLI images. As a result, the fiber orientation from a single 150µm SAXS pixel, for example, was compared to 50x50 SLI fiber orientations (px=3µm).”

b) the difference between SLI scatterometry and angular SLI.

SLI scatterometry measurements are very time and data consuming as they require to illuminate the sample from many different LED positions. As described in the Methods section, 80x80 LED positions were used for this purpose, yielding a series of 6,400 images to be evaluated. This type of measurement was only used to generate the scattering patterns in Figure 2C and compare them to the scattering patterns obtained from SAXS. All other figures were generated using angular SLI measurements, where the sample is illuminated around a circle with fixed polar angle, which is much faster and still sufficient to analyze the fiber orientations. As described in the Methods, 15°-steps were used, yielding a series of 24 images. The fiber orientations were computed from the positions of the peaks in the resulting azimuthal line profiles (similar to Figure 2C, bottom).

We have added a paragraph to the Methods section (“Scattered Light Imaging”) to make the difference between SLI scatterometry and angular SLI clearer and explain why we used two different methods:

“The SLI measurements were performed in two different ways: To generate the scattering patterns (upper Figure 2C), a time- and data-consuming scatterometry measurement was performed in which the sample was illuminated from 6,400 different angles, as described below. This was necessary to achieve sufficiently resolved scattering patterns for a visual comparison with SAXS scattering patterns. All other results were obtained from more time- and data-efficient angular measurements in which the sample was illuminated from 24 different angles around a circle, and the fiber orientations were derived from the peak positions in the resulting azimuthal profiles.”

c) how through plane orientations are computed from SLI.

We did not compute any through-plane fiber orientations from SLI measurements. We only computed through-plane fiber orientations from SAXS and dMRI measurements and compared those to the peak distances in the corresponding SLI line profiles (see Figure 6). Previous studies (Menzel et al., 2021a) suggest that the SLI peak distance is related to the through-plane fiber orientation. We validated this relationship for the first time with experimental ground-truth data from SAXS/dMRI.

We have added a sentence to the Methods section (“Generation of orientational parameter maps”) for clarification:

“The peak distance of the SLI azimuthal profiles was compared to the through-plane fiber orientations from 3D-sSAXS and dMRI to study the relationship between decreasing peak distance in SLI and increasing fiber inclination in SAXS and dMRI.”

d) how the correspondence of primary and secondary fibre populations was ensured across modalities.

In its current implementation, the SLIX algorithm that was used for all in-plane comparisons does not distinguish between primary and secondary orientations based on the strength of the peaks, but rather provides orientations as unit vectors. The first orientation is always the one which is closer to what is defined as 0° angle in the 0-360° range, the second orientation is the second closest, etc.

Here, we ensured the correspondence of matching first and second fiber populations across modalities when comparing in-plane fiber orientations in regions with crossing fibers: when computing the difference between in-plane fiber orientations from SAXS/SLI (Figure 4) and SAXS/dMRI (Figure 5—figure supplement 3), we always compared the first SAXS fiber orientation to both first and second SLI/dMRI fiber orientations, and used the one for which the angular difference was smallest. (“For each image pixel, the fiber orientations were subtracted…, taking the minimum of the two possible pairings in regions with crossing fibers”.) We then also computed the difference between the second SAXS orientation and the remaining SLI/dMRI orientation.

We have added a sentence to the newly named Methods section (“Image registration and pixel-wise comparison”) for clarification:

“In voxels with crossing fiber orientations, it was ensured that first and second orientation match across modalities, i.e. the first fiber orientation of one modality was compared to both orientations of the other modality, and the one with the smaller angular difference was used for comparison. The second orientation of the first modality was then compared to the remaining orientation of the second modality.”

Note that when comparing 3D fiber orientations (as in Figure 5E, and Figure 5—figure supplement 2) or when comparing through-plane fiber orientations (as in Figure 6), we only evaluated single fiber orientations from SAXS and dMRI-DTI analysis. As stated in the manuscript, this might lead to larger angular differences in regions with crossing fibers. This helped us identifying regions where SAXS and dMRI differ from each other, highlighting the good correspondence in most other regions.

4) The presentation of the study as a framework for combining SLI/SAXS/dMRI data, rather than a cross-validation study, caused some confusion. There are no clear prescriptions for how researchers interested in the neuroanatomical investigation would combine information from SLI and SAXS, or use the two methods in tandem, rather the information from one modality is used to corroborate/validate that from the other. Further, only SAXS is really compared to the diffusion MRI, presumably due to SLI having unreliable through-plane information. The authors will need to re-consider wording throughout the manuscript (see phrases such as "The combination of scattered light and X-ray imaging …[enable] detailed investigations of complex tract architecture in the animal and human brain", "The high-resolution properties of the former combined with the high-specificity of the latter enables the detailed reconstruction of multiple nerve fiber orientations for each image pixel, which can provide providing unprecedented insights into brain circuitry." or "The presented framework can … deliver more accurate brain connectivity maps of the animal and human brain." for example), or else additional information and example analyses should be provided on how these modalities can be combined for neuroanatomical investigation, particularly in regions where the information from the different modalities does not agree.

We thank the reviewers/editors for the valuable feedback and regret that our wording caused some confusion.

We understand that “framework” is not the appropriate word in this context, as it can raise false expectations. We have changed the wording throughout the manuscript, explaining that the current study demonstrates the experimental combination of the methods, and lays the groundwork for an eventual combined analysis in the future.

We have changed the wording throughout the manuscript, using “groundwork” instead of “framework”, and reformulated sentences in Abstract, Introduction, and Conclusion:

“[…] to achieve a *groundwork* for imaging axonal orientations”.

“The presented *SAXS and SLI results on the same specimens and their cross-validation provide the groundwork for an eventual combined analysis of the two techniques […].”.*

“We here *show* measurements of scattered *visual* light and X-ray scattering (SLI and SAXS) on the same brain tissue sample […]”.

It is also true that, while our study demonstrates experimental feasibility of combining the three methods on the same sample, it focuses on the comparison and cross-validation of the different techniques. We present parameter maps of the same tissue samples side by side and pixel-wise differences, but we do not provide any combined information e.g. in form of combined fiber orientation/vector maps. The reconstruction of crossing fibers has just recently been reported in SAXS and not been validated yet. SLI, on the other hand, has shown to yield crossing fibers with high accuracy and resolution, but the through-plane fiber orientations have not been validated yet. All techniques have their individual strengths and weaknesses so that a cross-validation (and future combination) is highly beneficial. Our study is the first necessary step (groundwork) towards a combination of SLI, SAXS and dMRI, which – in the long term – will lead to an enhanced interpretation and reconstruction of the brain’s nerve fiber architecture. When talking about a “combination“ of these techniques, we were referring to combining the *measurements*, i.e. enabling measurements on the same tissue sample, – and not combining the measurement *results*, i.e. providing combined analysis/parameter maps. The latter, while very much needed in the field, would require work beyond the scope of this manuscript, which we hope to perform with more samples in the future. Along these lines, we have removed the term “combined” throughout the manuscript.

We have removed mentions of “combined/combination” and changed “combined measurements of SLI and 3D-sSAXS” to “measurements of SLI and 3D-sSAXS on the same tissue sample” to avoid confusion. We have rephrased the sentences mentioned above and other phrases in Abstract and Conclusion:

“Here, we apply both techniques on the same samples and cross-validate them, laying the groundwork for providing ground-truth axonal orientations […]”.

“scattered light and X-ray imaging can provide quantitative micrometer 3D fiber orientations […]”.

“The high-resolution properties of the former and the high-specificity of the latter can both lead to reliable detailed fiber orientation maps.”

“By measuring the same tissue samples with SLI and 3D-sSAXS […]”.

In addition, we have added a new paragraph to the Discussion (“Towards a combination of SLI, SAXS, and dMRI”) about how the different parameter maps could be combined and what challenges we are facing:

“Our study mainly focused on the comparison and cross-validation of the different techniques. While enabling combined measurements of SLI, SAXS, and dMRI on the same tissue samples, we did not combine measurement results, e.g. in form of combined fiber orientation maps that integrate results from two or all three techniques. As all techniques have their individual strengths and weaknesses, and complement each other, a combined analysis of SLI/SAXS/dMRI measurement results would greatly enhance the accuracy and reliability of brain connectivity maps. However, before this becomes feasible, several challenges need to be faced: While dMRI provides volumetric information of thick tissue samples, SLI can only be performed on thin tissue sections. SAXS tensor tomography yields fiber orientations in larger tissue volumes, but the reconstruction of several crossing fibers per tissue voxel has so far only been demonstrated in tissue sections (Georgiadis et al., 2022). Robust tensor tomographic reconstruction of crossing fibers or providing quantitative out-of-plane angles via SLI would allow to fully combine the results; both developments are currently being pursued by the community. Another challenge would be to combine the fiber orientation vectors in each voxel, taking the different resolutions and accuracies of the techniques into account, for instance by fostering through-plane fiber orientations of SLI/SAXS by dMRI and verifying in-plane directions of dMRI by SLI or SAXS before using the data for tractography. We believe that tackling the issue of lack of ground truth in fiber orientation imaging is crucial, and requires synergies between the different methods. Hence, we imagine that – after combining the methods on multiple different samples, e.g., in the way demonstrated in this study – one can identify the data portions of highest confidence in each method, and use machine learning algorithms to improve the outputs of the other methods. Eventually, this could also improve dMRI and in-vivo applications.”

5) This may be an issue with the paper submission, but unfortunately the PDF figures were low resolution making the fibre orientations difficult to see. Higher-resolution images would be required in print.

We regret that the resolution of the figures was insufficient to properly see the fiber orientations. This must have occurred during the submission process, when generating the pdf version of the manuscript. The data shared with the manuscript were the original parameter maps; when opened e.g. with ImageJ, they can be zoomed in to see the individual pixels.

We have provided all figures as separate files with the original high-resolution images.

Reviewer #1 (Recommendations for the authors):This study presents a valuable comparison of fibre orientation estimates from three different modalities: diffusion MRI, scattered light imaging, and x-ray scattering. The comparison is interesting as each modality is sensitive to different aspects of tissue microstructure – water anisotropy, micron-scale structural coherence, and myelin lamella respectively. Where scattered light and x-ray imaging can be only applied ex vivo, diffusion MRI has in vivo applications but suffers from being an indirect estimate of the microstructure of interest. By acquiring all modalities in both a vervet monkey and human brain sample, the authors provide quantitative, pixel/voxel-wise comparisons of fibre orientation estimates within the same tissue samples. The authors show convincing agreement in fibre orientations from all three methods, giving confidence in the fidelity of the methods for neuroanatomical investigations. Differences are also observed: SLI is shown to have less reliable estimates of fibre inclination, and the CSD analysis presented overestimates the number of crossing fibre populations when compared to the microscopy methods, particularly in single fibre regions such as the corpus callosum, a known artefact in some diffusion analyses.In the current PDF, it is very difficult to see fibre orientations in figures due to low resolution, limiting the reader's ability to assess the results. Higher-resolution images would provide more information and easier comparisons.The methods are generally clear though some additional information is needed: 1) to specify the resolution that the orientations are compared in each figure and how data was up-/down-sampled for these comparisons respectively. For example, each SAXS pixel contains many SLI pixels. It is currently unclear whether the mean SLI orientation from a neighbourhood is equivalent to the SLI compared, or whether a comparison was made for each SLI pixel. Similarly, for the dMRI-microscopy comparisons. 2) I also could not follow why two SLI methods are presented in the methods: SLI scatterometry relating to Figure 2, and angular SLI relating to all other results. Further clarification is needed. 3) Since the quality of the data co-registration can strongly impact pixel/voxel-wise comparisons, quantification of the registration accuracy or overlays demonstrating the quality of the co-registration would be valuable.A primary weakness of the work as a diffusion MRI validation study is that though diffusion MRI supports many different models to extract fibre orientations with different outputs, here only a single model is compared to the microscopy data, which may affect the generalisability of the results. Further, it only compares the primary orientations from the diffusion MRI and does not consider each fibre population's magnitude (density of fibres) or the orientation dispersion, both of which can influence downstream analyses.The paper could be strengthened by a more detailed discussion on the differences between the imaging modalities – e.g. in terms of imaging resolution, signal-generating mechanisms, and sensitivity to specific aspects of the tissue microstructure – and how these differences may limit their application to specific neuroanatomical investigations, or ability to validate one another. For example, the microscopy sections are 80 microns thick whilst the diffusion voxel is 200 microns. I expect this could contribute to the difference in the number of fibre populations per voxel.The hypothesis that dMRI signal contributions from extra-axonal water result in additional fibre populations could be investigated by running CSD on both low and high-b-value data (for example using the openly available MGH dataset, Fan 2016) where fewer secondary fibre populations should be observed at high b-value.

We sincerely thank Reviewer #1 for the constructive feedback, which helped us to significantly improve our manuscript. We hope to have done our best to address all concerns:

First, we regret the insufficient resolution of figures. The resolution must have been reduced during the submission process, when generating the pdf version of our manuscript. We have now submitted all figures as separate files with the highest possible resolution. In addition, all parameter maps are publicly available and can be opened and zoomed in, e.g. with ImageJ, to see the fiber orientations of individual image pixels.

As requested by the reviewer, we have modified our manuscript and added additional methods information.

1) Concerning the data up-/downsampling: We have now specified in each figure caption at which resolution the images were compared and added the following explanation to the newly named Methods section “*Image registration and pixel-wise comparison*”: To minimize loss of information, the pixel/voxel-wise comparisons were performed at the spacing of the highest resolution image, i.e. the lower-resolution diffusion MRI (dMRI) and small-angle X-ray scattering (SAXS) images were upscaled to match the higher-resolution scattered light imaging (SLI) images. As a result, the fiber orientation of one SAXS pixel (px=150µm) was compared to the fiber orientations of 50x50 SLI pixels (px=3µm), and not to the mean; similarly for comparisons with dMRI.

2) Concerning the two SLI methods: We have added the following explanation to the Methods section “*Scattered Light Imaging*” to clarify why we used two different methods: To generate the scattering patterns (upper Figure 2C), a time-consuming SLI scatterometry measurement was performed in which the sample was illuminated from 6,400 different angles, as described in Menzel *et al.* (2021b). This was necessary to achieve sufficiently resolved scattering patterns for a visual comparison with SAXS scattering patterns. The fiber orientations can also be extracted from the peak positions in the azimuthal profiles (cf. bottom Figure 2C), without taking the overall shape of the scattering patterns into account. Therefore, all other results were obtained from more time- and data-efficient angular SLI measurements in which the sample was illuminated from 24 different angles around a circle and the fiber orientations were derived from the peak positions in the resulting line profiles, as described in Menzel *et al.* (2021a).

3) Concerning the quality of the co-registration: We thank the reviewer for this comment. We agree that the accuracy of image registration has a high impact on pixel/voxel-wise comparisons and determines the quality of our cross-validation study. We have added a new Discussion section “*Quality of cross-validation*” and inserted a new figure (*Figure 4—figure supplement 1*) to demonstrate the accuracy of image registration, both for the vervet and human brain samples: The reference and registered images are shown both in direct comparison (top and middle images, respectively) and as overlays (bottom images), as suggested by the reviewer. Reference and registered images show good correspondence (white/gray matter boundaries coincide). Only the fornix of the vervet brain section is not aligned (it moved when re-mounting the sample) so that this region was evaluated separately, as described in the manuscript. We found standard linear transformations (scaling, rotation, and translation) to be sufficient for achieving a fair comparison between the different modalities, demonstrating the experimental feasibility of our approach. There might still be individual voxels that were not sufficiently well aligned, especially when comparing sections (SLI/SAXS) to volumetric measurements (dMRI). However, this would only increase the angular differences between the fiber orientations. Our results can therefore be considered as an upper bound. Using standard linear transformations, we could already show that in-plane crossing orientations from SAXS and SLI, and through-plane orientations from SAXS and dMRI correspond very well to each other.

We understand the focus of our work lying rather on the cross-validation/evaluation of light and X-ray scattering, in comparison to dMRI which is much longer established, than on a “diffusion MRI validation study”: the myelin specific SAXS orientations and crossings were cross-validated with the high-resolution SLI orientations, and SLI out-of-plane fibers were validated using SAXS/dMRI as ground truth data.

The reviewer rightly noted that we used a single analysis method to extract fiber orientations from dMRI data (based on the MRtrix3 *dwi2response* and *dwi2fod* commands, using the dhollander and msmt_csd algorithms, respectively). Although to our knowledge this method is one of the most widely used for deriving fiber orientations for subsequent tractography, it is true that other methods might yield different results and that we cannot draw conclusions for diffusion MRI in general. We have included these considerations in the newly named Discussion section “*Comparison of SAXS and SLI fiber orientations to dMRI*”.

It is also true that our comparison focused on primary dMRI orientations without taking fiber density or dispersion into account. We decided to do so because deriving such metrics from SLI or SAXS data has not been implemented yet. However, we expect this to happen in the following years, enriching future studies. We have also included these aspects in the Discussion section.

We agree with the reviewer that our paper could be strengthened by a more detailed discussion on the differences between the imaging modalities. We have added a paragraph to the new Discussion section “*Quality of cross-validation*”: We compared results from three different imaging techniques (SLI, SAXS, dMRI) which all have different signal-generating mechanisms and resolutions. The different resolutions should be taken into account when interpreting the comparative studies. To investigate the relationship between SLI peak distance and fiber inclination, we used dMRI/SAXS images with at least 50 times lower in-plane resolution as reference (*Figure 6*). This is sufficient to validate the theoretical predictions, but insufficient to validate individual pixel values. To validate crossing fiber orientations from SAXS, we used SLI images with 30 times higher in-plane resolution, leading to a broad distribution of angular differences (depending on the region), but the mean difference around zero is evidence for a good overall correspondence (*Figure 4*). Finally, when comparing fiber orientations in SAXS and SLI to dMRI (*Figure 5*), it should be taken into account that dMRI voxels (with 200µm size) contain more fiber layers than the corresponding SAXS or SLI voxels (with 80µm section thickness), so that dMRI voxels might include additional fiber populations not present in SAXS or SLI data. On the other hand, fiber orientations that occur both in dMRI and SAXS voxels – like the out-of-plane fiber orientations from SAXS and dMRI (e.g. Figure 6B-C) – can be considered as reliable, given the substantially different contrast-generating mechanisms.

Finally, we thank the reviewer for the suggestion to study different *b*-values (last comment). We agree that an analysis based on different *b*-values might yield different results. Especially, an analysis with high *b*-values is expected to be more specific to the fiber orientations, as most other components of the signal would have already been attenuated. To investigate this hypothesis, we have run a separate analysis with high *b*-values only (5 and 10ms/μm^2^) and added a new supplementary figure (Figure 5—figure supplement 4) that compares the results for all *b*-values to high *b*-values only. We found that the fiber orientation distributions are almost identical between all *b*-values and high *b*-values only.

Thanks to the authors for providing links to the code and data relevant to the manuscript.It would be very interesting to have the authors comment on the ability of SLI/SAXS to validate dMRI estimates of fibre density of dispersion or demonstrate a comparison of these.

We thank the reviewer for this suggestion. This is indeed an important question.

We have tried to address this issue by a paragraph at the end of the newly named Discussion section “Comparison of SAXS and SLI fiber orientations to dMRI”:

“In the present study, only fiber orientations were compared, and not fiber dispersion/orientation distributions. This has not been implemented yet in SLI and SAXS, although it would be possible to derive this information, e.g. by a process similar to the constrained spherical deconvolution in dMRI where the signal from a homogeneous fiber bundle (fiber response function) is identified and deconvolved from the signal in other voxels to obtain the fiber orientation distribution for each voxel. Given the recent development of both methods, we expect this to be performed in the following years and enrich future studies.”

Some suggestions on the manuscript:– Abstract: "we find a reduction of peak distance with increasing out-of-plane fiber angles" – Without additional context from the paper, I found this statement unclear. Some further explanation or rephrasing would help.

We see that the statement is difficult to understand without additional context from the paper.

We have removed the detailed description in the Abstract and rephrased the sentence:

“We further use SAXS/dMRI to confirm theoretical predictions regarding SLI determination of through-plane fiber orientations.”

– Line 58 – Please can you give the resolutions of SLI/SAXS

The resolution of SLI depends on the magnification of the camera lens and sensor pixel size, while the resolution of SAXS is determined by the beam diameter and motor step size. For the current manuscript, SLI measurements were performed with an in-plane resolution of 3μm/px, and SAXS measurements with 100μm/px (vervet brain section) and 150μm/px (human brain section), as described in the Methods and figure captions.

As suggested, we have further specified the resolutions of SLI and SAXS in the Introduction, as given in the cited literature:

“Recent studies further revealed that SAXS can exploit the modulations in the azimuthal position of the myelin-specific Bragg peaks to resolve crossing nerve fiber populations across species *with resolutions of tens of micrometers* (Georgiadis et al., 2022)”

“SLI has been shown to reliably reconstruct up to three in-plane fiber orientations for each image pixel (with an accuracy of +/-2.4°; Menzel et al., 2021a) *and an in-plane resolution at the micrometer scale (Menzel et al., 2021b).*

In addition, we have specified the resolution dependencies of SLI and SAXS in the Methods section:

“the in-plane resolution is determined by the magnification of the camera lens and the sensor pixel size”(SLI).

“the in-plane resolution is determined by beam diameter and step size” (SAXS).

– Line 123 "Figure 4A shows very small angular differences that appear to be uniformly distributed" – I'm not sure what is meant here the distribution doesn't appear to be shown.

We realize that we expressed ourselves somewhat misleadingly. We meant the distribution of regions with positive/negative angular differences (red/blue) across the tissue, and not a distribution of values (as the histogram in Figure 4D). When saying that “small angular differences … appear to be uniformly distributed” we meant that the positive and negative angular differences (displayed in shades of red and blue in Figure 4A) appear to be uniformly distributed across the tissue, i.e. there is no visible pattern or a clear over-/under-estimation in certain regions.

We have changed the wording accordingly:

“Figure 4A shows that the small positive and negative angular differences (displayed in shades of red and blue, respectively) appear to be randomly distributed across the tissue.”

– Figure 4 – please can you clarify why some regions do not appear to be included in the analysis? i.e. the top right-hand side of the slide is missing in A/B and parts of the cingulum appear to be missing in part C. What is the rationale for this? By excluding many pixels from the cingulum in the histogram in 4d, would this not lead to an underestimation of the angular difference?

The reviewer is rightfully confused by the apparent exclusion of some areas. Regarding the top right region, the sample was slightly rotated when performing the SAXS measurement, resulting in a non-overlapping area in the top right corner after registering SAXS onto SLI (see also new Figure 4—figure supplement 1A). This region appears black in Figure 4A/B and uncolored in C.

We have made the top right-hand corner black in all three images (Figure 4A-C) to illustrate that this region was excluded after registration.

Regarding the cingulum, this appears partly missing in Figure 4C, because, as mentioned in line 120 in the first version of the manuscript, we only compared pixels in which both techniques yield the same number of fiber orientations. Regions with highly inclined fibers, such as the cingulum, yield rather flat azimuthal profiles with multiple small peaks (cf. Figure 2(iv)). This is why, for many pixels in this region, no reliable fiber orientation could be determined for SAXS and/or SLI, and no angular difference could be computed. As mentioned at the beginning of the Results section, “the information about the in-plane fiber orientations is limited” in such regions. Another such region where there are more voxels with highly inclined fibers and where the angular difference could not be computed for all image pixels is the mid-right end of the field of view, which contains fibers of the superior longitudinal fasciculus.

We have added a more detailed explanation to the new section “Quality of cross-validation” in the Discussion:

“For pixel-wise comparisons as in Figures 4 and 6, only image pixels in which both techniques yield the same number of fiber orientations were considered. To ensure that the determined in-plane orientations are reliable, they were only computed if the azimuthal profiles showed one/two dominant peaks (indicative of a single fiber population) or two dominant peak pairs (indicative of two in-plane crossing fiber populations), see Methods. Regions with inclined crossing or highly inclined fibers yield rather flat azimuthal profiles with multiple small peaks (cf. Figure 2(iv)), so that fiber orientations cannot be compared reliably. As a result, such pixels, e.g. at the upper part of the cingulum, were excluded from the pixel-wise comparison (see Figure 4A-C). In addition, in-plane orientations calculated in regions with highly inclined fibers are naturally less reliable (the projection onto the plane is much smaller) and were therefore at this point excluded from the in-plane analysis.”

– Figure 4a – I would recommend using a colour bar where colours have a unique meaning. White or light red currently can mean more than one thing in the image.

We thank the reviewer for this recommendation.

We have changed the color bar in Figure 4A (dark blue < -45°, white = 0°, dark red > 45°).

– Figure 5 – The 3D colour wheel is atypical for dMRI community which usually uses blue for IS and green for AP. Though the current version is acceptable as the meaning of the colours is clearly indexed, it may be worth considering changing to avoid any confusion.

We thank the reviewer for this comment. We agree that the use of a different 3D color-coding might be confusing for the dMRI community.

We have changed the 3D color-coding in “Figure 5C,D,F” and “Figure 5—figure supplement 2” as suggested (x=red, y=blue, z=green).

– Line 173 onwards – Please can you provide the maps of single/crossing fibre voxels for both dMRI and SAXS? It would be interesting to see the clustering of the single/crossing fibre populations.

We thank the reviewer for this suggestion.

We have now added a supplementary figure (Figure 5 - figure supplement 5) that shows voxels with single and crossing fiber populations in cyan and yellow, respectively.

– Line 203 – I find it very hard to interpret or see the 3D arrows in Figure 6A

The 3D arrows indicate the orientation in which the fibers point out of the section plane (independent of their inclination, which is shown in different colors).

We have modified the wording and enhanced the boundaries of the 3D arrows in Figure 6A so that they are hopefully now better to see.

– Line 413 – "The high-resolution properties of the former combined with the high-specificity of the latter enables the detailed reconstruction of multiple nerve fiber orientations for each image pixel, which can provide providing unprecedented insights into brain circuitry." This conclusion seems out of place given the work presented, as details have not been provided on how to combine information from these modalities.

We performed combined measurements on the same tissue sample, but do not provide any combined information from the different measurement results. Therefore, we understand that this conclusion does not reflect our study’s content, but rather the end goal of combining the modalities. We have added a paragraph to the Discussion (“Towards a combination of SLI, SAXS, and dMRI”) to discuss how information from the different modalities could be combined, and reformulated the sentence. (→ Please see our response to “Essential Revisions” 4.)

– Line 414 – Typo "provide providing"

We thank the reviewer for the careful proofreading.

We have corrected this typo.

– Line 429 – I assume the tissue was immersion fixed?

The tissue was indeed perfusion- and immersion fixed.

We have added this information to the Methods section:

“The brain was perfusion-fixed with 4% paraformaldehyde […], immersed in 4% paraformaldehyde for several weeks […]”

Reviewer #2 (Recommendations for the authors):This work is a cross-validation of an x-ray tomography technique (SAXS) and an optical microscopy technique (SLI) for imaging axonal orientations ex vivo. These innovative methods were introduced in recent papers by the authors, who have teamed up here to compare them side-by-side on the same tissue samples for the first time. The two methods are both label-free (do not require staining) and they are quite complementary. SAXS can provide full 3D orientation measurements on intact tissue, but it operates at a mesoscopic resolution and requires access to a synchrotron. SLI can measure the orientations of multiple fascicles per voxel at a microscopic resolution and relies on more widely accessible equipment, but its accuracy suffers for fiber orientations perpendicular to the imaging plane and it requires tissue to be sectioned before it is imaged. Therefore it makes a lot of sense to explore the complementary strengths of these two techniques, and to use one to "fill in the blanks" of the other. The paper also compares the orientation measurements obtained with SAXS and SLI to those obtained with diffusion MRI. The latter provides only indirect measurements based on water diffusion, at a mesoscopic resolution somewhat lower than that of SAXS, but has the benefit of being feasible in vivo.A limitation of this study is that conclusions on the comparison between SAXS and SLI are drawn from only 2 sections of a partial monkey brain sample and 2 sections of a partial human brain sample. Conclusions on diffusion MRI are drawn only on the 2 human sample sections. This is particularly an issue for the comparison to diffusion MRI, as the diffusion MRI voxels are wider than the section thickness, hence one cannot preclude that any orientations detected with diffusion MRI but not with SAXS and SLI come from the portion of the voxel that is missing from the corresponding SAXS/SLI section.The stated aim of the paper is to provide a framework for combining the complementary benefits of SAXS and SLI, rather than simply presenting the results of a cross-validation study. This is a significant and ambitious aim. However, in order for this to serve as a framework, there would have to be clear prescriptions for how researchers interested in obtaining ground-truth measurements of axonal orientations would do so by using these two methods in tandem. This is not adequately developed in the paper in its present form. For example, the results show reasonable agreement between SAXS and SLI orientations when fibers lie within the SLI imaging plane and decreasing agreement for fibers with increasing through-plane inclination. How would the two methods be combined in voxels where they disagree? Would one use SLI orientations in voxels with fewer through-plane fibers and SAXS orientations in voxels with more through-plane fibers? How would voxels be assigned to each category? How would the orientation vectors from the two modalities be composed and how would the resolution difference between the two be handled? When the through-plane measurement of SLI is unreliable, is its in-plane measurement still reliable? That is if there were one mainly in-plane and one mainly through-plane fiber population, would the orientation of the former still be measured correctly by SLI? There is also considerable agreement reported here between through-plane orientations obtained with SAXS and diffusion MRI. Would this mean that diffusion MRI itself could be used to supplement SLI with through-plane orientations? Any clear set of prescriptions along these lines would represent a framework for imaging orientations by combining modalities. This, however, would require detailed steps for how to perform the combination and use the multi- vs. uni-modal framework to reconstruct connectional anatomy.A key advantage of SAXS is that it can be performed on intact samples, i.e., before any nonlinear distortions of the tissue are introduced by sectioning. Thus it can provide an undistorted reference, with contrast on axonal orientations that would be absent in, say, a structural MRI of comparable resolution. This contrast could be used to drive registration of the distorted SLI sections to an undistorted SAXS volume, and therefore is a key way in which the two techniques can complement each other. Here, however, this is not explored, as SAXS is performed after sectioning. It is not clear if this is the authors' prescription for how a combined SAXS/SLI framework would be implemented, or if it was done specifically for this study. First, it would seem that SAXS on the intact sample would be lower maintenance, requiring less setup time and hence potentially less overall beamtime than performing SAXS on each section separately. This would make it more practical for routine deployment beyond a few sections. Second, because the SAXS data are now nonlinearly distorted, they cannot be affinely aligned to the MRI volumes. While, in principle, performing both SAXS and SLI on the sections may facilitate the comparison between the two, having to unmount, rehydrate, and remount the sections in between may negate this advantage, as now there is no guarantee that SAXS and SLI can be affinely registered to each other. Here all these registration steps are performed affinely, so it is unclear to which extent the computed errors between modalities are characterizing the inherent limitations of the respective contrasts, or limitations of the registration technique. Some of the alignment is performed manually, for example, specific regions of the images are realigned by hand, and the slice of the diffusion MRI volume that is aligned to the SAXS/SLI sections is chosen by hand. Again, for this to serve as a framework that can be deployed on whole samples, there would have to be clear prescriptions for how to perform these steps robustly, how to ensure that the MRI can be acquired in a coordinate frame parallel to the sections, etc.Finally, the paper puts forth a general conclusion that diffusion MRI overestimates the number of fiber populations per voxel, on the basis of small ODF peaks appearing perpendicular to the main ODF peaks. Of all conclusions in the paper, this is the least convincingly supported by evidence. First, these small perpendicular peaks are a known artifact, which would be typically eliminated by ignoring ODF peaks below a certain amplitude, a common practice in diffusion tractography algorithms. The authors refrain from using an amplitude threshold, with the rationale that it may also remove true diffusion orientations. However, they apply a threshold when they detect SLI peaks (a rather stringent 8% of the maximum). Second, the explanation that these artifactual peaks may appear due to vessel walls is not convincing. Vasculature is sparse. A single vessel wall will not impact the diffusion signal in the same way as a bundle of parallel axons. In an axon bundle, water molecule displacements are restricted in all directions except parallel to the axons. A single vessel wall in a voxel will not have the same effect on displacements (which are much smaller than the size of the voxel). From Figure 5, it looks like there would be at most 1-2 of these vessels in a diffusion MRI voxel, and they would not be in all voxels. This cannot explain the widespread appearance of these small artifactual peaks. Third, many ODF reconstruction methods have parameters that can be adjusted to make these artifactual peaks more or less prominent. The default parameters may be optimal for in vivo but not ex vivo data, due to the effects of fixation. In light of these concerns, I would caution against making such a general statement about all diffusion MRI in the human brain, especially on the basis of a single diffusion reconstruction method applied to a single location in one brain.

We sincerely thank Reviewer #2 for the constructive feedback, which helped us to significantly improve our manuscript. We hope to have done our best to address all concerns:

First, regarding the limited number of tissue sections used for our study (second paragraph):

It is true that we only evaluated a limited number of samples – mainly due to the limited beam time available for SAXS experiments. We believe that the main conclusions concerning the cross-validation of SAXS crossing fibers and SLI out-of-plane fibers still remain valid.

The reviewer correctly points out that the dMRI voxels (with 200um size) are wider than the section thickness (80um) so that additional fiber orientations detected with dMRI might come from the portion of voxels missing in the corresponding SAXS/SLI measurement. We have added a clarifying paragraph in the newly named Discussion section “*Comparison of SAXS and SLI fiber orientations to dMRI*” as well as in the new Discussion section “*Quality of cross-validation*”. Nevertheless, we do not expect additional fiber orientations in comparable homogeneous regions like the corpus callosum, and fiber orientations that occur both in dMRI and SAXS/SLI – like the out-of-plane fiber orientations from dMRI and SAXS (e.g. *Figure 6B-C*) – can be considered as reliable, given the substantially different contrast-mechanisms of the microscopy and dMRI techniques.

Concerning the aim of our paper and the questions raised by the reviewer in the third paragraph:

We understand that the term “framework” is not the appropriate word in this context, as it can raise false expectations. Our aim was rather to provide a basis (“groundwork”) to enable combined measurements of SLI/SAXS (and dMRI) on the same tissue samples and cross-validate the techniques (the crossing fiber orientations in SAXS and the through-plane fiber orientations in SLI have not been validated using other techniques so far). We have changed the wording throughout the manuscript, explaining that we focused on laying the “groundwork” instead of providing a “framework”, and reformulated the corresponding sentences.

Our aspiration was to provide a protocol how the complementary imaging techniques can be performed on the same tissue sample. When talking about a “combination” of techniques, we were referring to combined measurements (i.e. measurements on the same sample), and not to a combined analysis (e.g. in form of combined parameter maps and fiber orientation vectors). The latter, while very much needed in the field, would require many more and heterogeneous samples, and work beyond the scope of this manuscript, which we hope to perform in the future. Along these lines, we have removed the term “combined” throughout the manuscript, and wrote e.g. “measurements of SLI and 3D-sSAXS on the same tissue sample” instead of “combined measurements of SLI and 3D-sSAXS” to avoid confusion.

However, it is of course a valid question how SAXS and SLI can be combined in voxels where they disagree, how the orientation vectors can be composed, and how the resolution difference between the methods can be handled. We have added a new Discussion section “*Towards a combination of SLI, SAXS, and dMRI*” to elaborate on how a combined analysis (e.g. in form of combined fiber orientation maps) can be achieved and what challenges we are facing.

Concerning the reviewer’s question if the orientation of an in-plane fiber population would be correctly measured by SLI if there was another through-plane fiber population: We only evaluated regions belonging to a single fiber population (SLI azimuthal profiles with one or two dominant peaks) and regions belonging two in-plane crossing fiber populations (SLI azimuthal profiles with two dominant peak pairs). Voxels containing both in-plane and through-plane fibers were excluded from the analysis. The determined in-plane SLI orientations can thus be considered as reliable. We have added these aspects to the new Discussion section “*Quality of cross-validation*”.

Regarding the reviewer’s question if dMRI itself could be used to supplement SLI with through-plane orientations: Diffusion MRI could indeed be used as a reference to enhance the interpretation of through-plane fiber orientations from SLI measurements. One disadvantage over SAXS is the lower resolution and that it cannot directly be performed on the same tissue section as SLI. These aspects have also been added to the new Discussion section.

Concerning the reviewer’s suggestion to perform SAXS before sectioning and the problem of image registration (fourth paragraph):

It is true that SAXS tensor tomography can be applied to larger tissue volumes and that it is not limited to tissue sections. However, the reconstruction of crossing fibers has so far only been realized in sections (Georgiadis et al., 2022) and not in intact samples. As we wanted to cross-validate these fiber crossings using SLI as reference, we decided to perform the SAXS measurements on the same tissue sections as the SLI measurements. A comparison to results from SAXS tensor tomography might still be interesting in the future. We have added these considerations to the new Discussion section “*Towards a combination of SLI, SAXS, and dMRI*”.

It is also true that cutting a section from a brain tissue sample might introduce non-linear distortions; in particular, it is challenging to identify this particular section in the original tissue volume; unmounting and remounting of an already existing section introduces much less distortions. We have added a new figure (*Figure 4—figure supplement 1*) which shows that a co-registration with linear transformations (scaling, rotation, and translation) is already sufficient to allow for a fair comparison between the different image modalities, both for vervet and human brain samples. Only the fornix of the vervet brain section moved during remounting of the sample, and was therefore evaluated separately, as described in the manuscript. In any case, even if the angular differences in some image pixels were larger due to an imperfect co-registration, a perfect co-registration would only yield even smaller differences. Hence, the reported angular differences can be considered as upper bound, demonstrating that SAXS and SLI fiber orientations show already a very good correspondence. We have added a corresponding paragraph to the new Discussion section “*Quality of cross-registration*”.

Finally, we agree that a clear prescription would be necessary to enable combined analysis on whole tissue samples. As mentioned further above, our aim was to provide the groundwork for combined measurements on the same tissue sample and cross-validate the different techniques, and not to provide combined fiber orientation maps or similar. We have added our thoughts on how to combine the different image modalities to the new Discussion section “*Towards a combination of SLI, SAXS, and dMRI*”.

Concerning the final concern of the reviewer that an overestimation of the number of fiber populations per voxel is not sufficiently supported (last paragraph):

We understand this concern and have removed all phrases that could be understood as generalized claims for MRI, including any reference to fiber orientations overestimation. Furthermore, we have extended the Discussion to indicate the non-generalizability of our results.

Regarding the first point that the minor perpendicular ODF peaks could be removed by applying a suitable amplitude threshold: This is a valid remark and was discussed partly in the first version of the manuscript, when referring to increasing the threshold of secondary lobes prior to running tractography algorithms and to the problem that it might decrease the sensitivity for the cases where there exist actual but less prominent secondary fiber populations. We have extended the Discussion to address the concerns of the reviewer.

Regarding the second point that the minor ODF peaks are probably not caused by vessel walls: We thank the reviewer for the valid remarks and have removed all mentions of blood vessels in the manuscript, including the arrows in Figure 5H.

Regarding the third point that parameters can be adjusted to make the artifactual peaks more/less prominent, and that default parameters might be optimal for in vivo but not ex vivo data: We have added the remark that model parameters can be fine-tuned to decrease the percentage of false-positives to the Discussion.

Finally, it is true that we only used a single diffusion reconstruction method and measured only a single location in one human brain with dMRI. As mentioned at the very beginning, the number of samples was limited, and we included the reviewer’s concerns in the newly named Discussion section “*Comparison of SAXS and SLI fiber orientations to dMRI*”. For the main purposes of the paper like the cross-validation of out-of-plane fibers in SAXS/SLI, the dMRI data was still sufficient as we could show a good correspondence between dMRI/SAXS in these regions.

– The term "neuronal orientations" (or "trajectories") is used throughout but "axonal orientations" would be more suitable, as neurons are not imaged here.

We agree with the reviewer that the term “axonal orientations” is more suitable.

We have changed the wording throughout the manuscript and used “axonal” instead of “neuronal” orientations.

– "… the signal is affected by all brain structures, not axons". It is true that the diffusion signal is affected by all tissue components, but that is not the main issue. What we use to extract orientations is not the signal itself, but the anisotropy (orientational dependence) of this signal. Not all tissue components contribute to this anisotropy the same way that myelin sheaths and axon membranes do, even if they contribute to the signal itself.

We agree that the anisotropic component of the signal is certainly not affected by all brain structures.

We have changed the sentence “… the signal is affected by all brain structures, not only axons” to:

“… isolating the anisotropic signal coming from myelinated axons alone is challenging”.

– There are several mentions of false positives as the problem with diffusion tractography methods. However no method has a fixed false positive rate, all methods have thresholds that can be adjusted to make their false positive rate as low or as high as one wants. It's the trade-off between the false positive rate and true positive rate that is difficult to improve, not the false positives themselves. You can always remove false positives, but the challenge is that you cannot do it without an excessive loss of true positives.

The reviewer correctly points out that the false positive rate depends on the specific parameters used in each method, and that finding a balance between false positives and false negatives is challenging. We have addressed this issue in multiple ways.

We have removed most mentions of “false-positives” from the manuscript, especially in cases where we interpret our own results: We renamed the subsection “Identification of false-positive fiber tracts in dMRI” to “Comparison of SAXS and SLI fiber orientations to dMRI”, deleted a corresponding sentence in the Introduction (“Especially notable is that structural connectivity and wiring diagrams of the brain … contain a large percentage of false-positive fiber tracts …”) and also deleted the reference to false positive crossings in dMRI in the Conclusion. In the Discussion, we mention this fine balance: “A possible solution would be to fine-tune the model parameters so that the percentage of false-positives decreases”.

(Please also see our response to “Essential Revisions” 4.)

– In Methods: the in-plane resolution of the SAXS and SLI scans is provided, but what is the through-plane resolution? Is it the same as the section thickness (80 microns)?

It is indeed true that the through-plane resolution of SAXS and SLI corresponds to the section thickness (60µm for the vervet and 80µm for the human brain samples).

We have added this information to the corresponding Methods sections:

“While the in-plane resolution is determined by the magnification of the camera lens and the sensor pixel size, the through-plane resolution is limited by the thickness of the brain tissue section (60µm for vervet and 80µm for human).”

“While the in-plane resolution is determined by beam diameter and step size, the through-plane resolution is determined by the thickness of the tissue section (60μm for vervet and 80μm for human).”

– In Methods: there is a description of how the in-plane and through-plane orientations are computed in SAXS, but only how the in-plane orientations are computed in SLI.

(Please see our response to “Essential Revisions” 3c.)

– In Methods: it is unclear at what scale the angular errors are computed. Is it at the highest resolution among the modalities that are being compared, i.e., do you obtain as many values of the error as SLI-sized voxels?

The angular errors were indeed computed at the highest resolution among the different modalities, yielding as many error values as SLI-sized voxels. The lower-resolution images were upscaled to the higher-resolution images before comparison. (→ Please see our response to “Essential Revisions” 3a.)

– The use of the word "peak" without further qualifiers is likely to confuse the diffusion MRI audience. Here 2 peaks (in the SLI profile) = 1 fiber population, but in usual diffusion terminology 1 peak (in the diffusion/fiber ODF) = 1 fiber population.

We thank the reviewer for this comment. It is good to know that our wording might lead to confusion in the dMRI community.

We have clarified the definition:

“In the following, the term ‘peak’ will be used to refer to peaks in azimuthal profiles, so that a pair of azimuthal ‘peaks’ always corresponds to a single fiber orientation.”

– How are the "primary" and "secondary" orientations in a voxel determined for each modality? Are any steps taken to ensure that they match across modalities? Otherwise, you can imagine a scenario where the same fiber population ends up being the primary orientation in one modality but the secondary one in the other, and vice versa. This would overestimate the angular error between modalities. If the goal is to measure error as a difference between orientation angles, not amplitudes (as amplitudes mean different things in each modality anyway), then care must be taken to establish the correspondence of fiber populations between modalities.

We thank the reviewer for pointing this out. We decided indeed to make a comparison of orientation angles rather than amplitudes, as they mean different things in each modality. We ensured that first and second orientations match across modalities. (→ Please see our response to “Essential Revisions” 3d.)

– Registration with 12 degrees of freedom is affine. Linear and affine are not quite synonyms.

We thank the reviewer for the careful reading. We used indeed linear registration with 9 degrees of freedom (scaling, rotation, translation), and not – as stated in the first version of the manuscript – 12 degrees of freedom. We apologize for this mistake.

We have changed the description in the Methods accordingly (9 degrees of freedom, instead of 12).

– It would be helpful to report the angular resolution of each technique as a function of through-plane inclination, e.g., what is the smallest angle between crossing fibers that can be accurately resolved by SLI when the fibers are in-plane or inclined?

In general, it is more difficult to accurately determine the in-plane fiber orientation if the fibers are highly inclined, as could be seen in the cingulum (cf. Figures 3 and 4). In regions with in-plane and moderately inclined fibers, the determined in-plane fiber orientations can be considered as reliable. A function between angular resolution and through-plane fiber inclination cannot be given in the present or previous studies, and would probably require dedicated studies with phantoms. Please note that crossing fibers in SLI were only evaluated when the azimuthal profiles showed two peak pairs (peaks lying approx. 180° apart). Thus, regions with highly inclined crossing fibers (where peaks lie closer together) were not evaluated. The evaluation of inclined crossing fibers was not the focus of this study and will be left for future investigations.

We have added these aspects to the new Discussion section “Quality of cross-validation”:

“Another important aspect to note is that our cross-validation study focused on in-plane (single or crossing) fibers and through-plane (single) fiber populations. At this point, we cannot make a confident assessment about highly inclined, crossing fibers.”

“in-plane orientations in regions with highly inclined fibers are naturally less reliable (the projection onto the plane is much smaller) and should therefore at this point be excluded from the in-plane analysis”.

– In Discussion: the paragraph on clearing and fluorescence microscopy should be updated to reflect the latest developments in that field. It is not "only feasible for smaller sample sizes", as clearing has now been demonstrated in samples as large as a whole slab of a human hemisphere. Also, the statement "it fails to disentangle densely packed nerve fibers" is not generally true, as this depends on the imaging resolution. The resolution of fluorescence microscopy can in principle be increased to the sub-micron level. It is true that the few studies that compared fluorescence microscopy of cleared tissue to diffusion MRI so far did not use state-of-the-art of these methods, but this is not due to an inherent limitation of the methods.

We thank the reviewer for this clarification.

We have removed the sentence (“Moreover, it is only feasible for smaller sample sizes […] and it fails to disentangle densely packed nerve fibers.”).

– In Discussion: while a single PS-OCT measurement gives only the in-plane orientation, the use of multiple measurements at multiple incidence angles to infer the through-plane orientation has been demonstrated in the literature.

We thank the reviewer for this clarification.

We have removed the phrase (“but just as PS-OCT, the techniques only derive 2D fiber orientations”).

– In Discussion: the statement that SAXS and SLI yield orientations with "a higher precision and smaller crossing angles" should be clarified. Higher/smaller than what? Which modality here is used as the reference that SAXS, SLI, and diffusion MRI are being compared to, in order to determine that the error is lower in one than the other?

We agree that the statement is vague and should be rephrased.

We have changed the wording to:

“SAXS and SLI have both shown the potential to reliably resolve secondary (crossing) fiber orientations”